# ENHANCING MULTIMODAL LLM FOR DETAILED AND ACCURATE VIDEO CAPTIONING USING MULTI-ROUND PREFERENCE OPTIMIZATION

## ABSTRACT

Videos contain a wealth of information, and generating detailed and accurate descriptions in natural language is a key aspect of video understanding. In this paper, we present video-SALMONN 2, an advanced audio-visual large language model (LLM) with low-rank adaptation (LoRA) designed for enhanced video (with paired audio) captioning through directed preference optimization (DPO). We propose new metrics to evaluate the completeness and accuracy of video descriptions, which are optimized using DPO. To further improve training, we introduce a novel multi-round DPO (mrDPO) approach, which involves periodically updating the DPO reference model, merging and re-initializing the LoRA module as a proxy for parameter updates after each training round (1,000 steps), and incorporating guidance from ground-truth video captions to stabilize the process. To address potential catastrophic forgetting of non-captioning abilities due to mrDPO, we propose rebirth tuning, which finetunes the pre-DPO LLM by using the captions generated by the mrDPO-trained model as supervised labels. Experiments show that mrDPO significantly enhances video-SALMONN 2's captioning accuracy, reducing global and local error rates by 40% and 20%, respectively, while decreasing the repetition rate by 35%. The final video-SALMONN 2 model, with just 7 billion parameters, surpasses leading models such as GPT-4o and Gemini-1.5-Pro in video captioning tasks, while maintaining competitive performance to the state-of-the-art on widely used video question-answering benchmark among models of similar size. Upon acceptance, we will release the code, model checkpoints, and training and test data. Demos are available at https://video-salmonn-2.github.io.

## 1 INTRODUCTION

Large language models (LLMs) have exhibited outstanding capabilities in a wide range of natural language processing (NLP) tasks, and in some instances, have even approached human-level performance (OpenAI et al., 2024; Dubey et al., 2024; Touvron et al., 2023; Du et al., 2022; Bai et al., 2023a). LLMs' remarkable ability to understand, generate, and reason with text has sparked widespread interest among researchers, attracting both academia and industry to extend them to multimodal understanding and generation. To endow LLMs with multimodal understanding capability, recent studies adopted a paradigm of training modality adapters and aligners between multi-modal encoders and LLMs. This approach leverages world knowledge in the textual LLM to interpret diverse types of data perceived by multimodal encoders, enabling the generation of meaningful insights. Over the past two years, many multimodal LLMs have emerged following this paradigm across different modalities. These include models for image and silent video understanding (Liu et al., 2024b;a; Li et al., 2023; Bai et al., 2023b; Lin et al., 2023; Chen et al., 2023; Lin et al., 2024; Chen et al., 2024), audio understanding (Wu et al., 2023; Tang et al., 2024b; Chu et al., 2023; 2024; Gong et al., 2024; 2023; Tang et al., 2024c; Zheng et al., 2024), and audio-visual understanding (Team et al., 2024; Cheng et al., 2024; Sun et al., 2024; Fu et al., 2024b; Tang et al., 2024d).

Text descriptions of multimodal data are critical for building multimodal LLMs. This is because most contemporary multimodal LLMs treat multimodal captions as a cornerstone task during pre-training or supervised fine-tuning (SFT), to align the representation spaces of multimodal encoders with that of textual large language models, helping LLMs recognise and understand events in mul-

timodal data. Thus, collecting high-quality text descriptions paired with multimodal data is crucial for constructing high-performance multimodal LLMs, which implies training the model with more detailed and less hallucinated labels aligned with the multimodal data that could enhance the LLM's ability to perform multimodal understanding and reasoning. In video understanding, generating detailed and accurate captions is crucial but challenging, as videos contain rich content that encompasses not only spatial features within individual visual frames but also audio-visual events that unfold across multiple frames over time. However, very few multimodal LLM-related works focus on improving the quality of video captions, due to the lack of quantitative metrics for evaluating video captions and the absence of training methods to enhance the completeness of these descriptions while reducing the risks of hallucination. Additionally, while audio is typically paired with video and provides crucial, complementary information to the visual content, most current visual LLMs lack audio-understanding abilities, leading to the omission of audio information in the generated captions.

In this paper, we introduce video-SALMONN 2, a multimodal LLM that supports both audio and visual inputs and primarily focuses on detailed and holistic audio-visual captioning. Building upon an already well-trained visual LLM, video-SALMONN 2 is further enhanced with auditory capabilities by training on audio-only data as well as videos with synchronized audio tracks. This enables the model to simultaneously "see" and "hear" the video, emulating the way humans perceive and interpret multimedia content. To accurately assess the performance of the model, new metrics to evaluate captioning quality are proposed, which then serve as the objective to optimize during reinforcement learning (RL) based on direct preference optimization (DPO) (Rafailov et al., 2024). A novel multi-round DPO (mrDPO) is proposed and performed based on the preferences guided by the metrics, followed by a novel rebirth tuning stage to avoid the degradation of the non-captioning abilities caused by the mrDPO. The rebirth tuning leverages the post-mrDPO model to revise the captions of the videos in the training set, and trains the model after audio modality alignment using supervised fine-tuning (SFT) with the revised training data. Experiments demonstrate that video-SALMONN 2 with 7 billion (B) parameters can generate complete and accurate video descriptions and even outperforms much larger commercial multimodal LLMs such as GPT-4o and Gemini-1.5-Pro, and it also maintains competitive performance to the state-of-the-art (SOTA) multimodal LLM of similar model size on commonly used video question-answering (QA) benchmarks including Video-MME (Fu et al., 2024a), NeXT-QA (Xiao et al., 2021), MVBench (Li et al., 2024a) and VideoVista (Li et al., 2024b).

The main contributions of this work can be summarized as follows:

- We develop video-SALMONN 2, a powerful audio-visual LLM that generates high-quality video captions, outperforming larger commercial models such as GPT-4o and Gemini-1.5 in terms of completeness and accuracy.
- We introduce an evaluation pipeline that computes the missing and hallucination rates of audio-visual events in video captions using text-based LLMs, breaking down the process into sub-tasks suited for current LLMs. Additionally, we provide a new benchmark for video captioning with a human-annotated test set.
- We propose the mrDPO approach to optimize multimodal LLMs for video captioning, incorporating periodic updates to the DPO reference model, merging and reinitializing the low-rank adaptation (LoRA) (Hu et al., 2022) module, and smoothing the training loss using SFT based on ground-truth captions. To our knowledge, this is the first work applying RL to audio-visual LLMs.
- We introduce rebirth tuning to ensure the resulting model maintains high performance in both captioning and non-captioning tasks. The mrDPO process, followed by rebirth tuning, can be iteratively applied to further enhance performance.

## 2 RELATED WORK

### 2.1 MULTIMODAL LLMS

Following the paradigm of connecting multimodal encoders to LLMs using modality adapters, various models have been developed. For image-based LLMs, LLaVA (Liu et al., 2024b;a) applies instruction tuning (Wei et al., 2022) to enhance performance on zero-shot tasks. BLIP-2 (Li et al., 2023) uses Q-Former to link a frozen encoder with an LLM, while VILA (Lin et al., 2023) explores

pre-training strategies, achieving impressive results in video QA. InternVL (Chen et al., 2023) scales up the size of visual encoders for improved image representation. For silent video understanding, Video-LLaVA (Lin et al., 2024) aligns both image and video adapters to learn unified representations. ShareGPT4Video (Chen et al., 2024) uses GPT-4 to generate dense video captions, improving data quality, and LLaVA-Hound (Zhang et al., 2024) introduces DPO to enhance video LLMs' understanding capabilities.

In the realm of audio perception, SALMONN (Tang et al., 2024b) uses a dual-encoder structure and can perform zero-shot audio reasoning tasks. LTU (Gong et al., 2024) and LTU-AS (Gong et al., 2023) trained on a large audio question-answering dataset can answer open-ended questions about audio. Qwen-Audio (Chu et al., 2023) and Qwen2-Audio (Chu et al., 2024) are built on large amounts of audio data to achieve high performance on a wide range of carefully selected audio tasks. Zheng et al. (2024) and Tang et al. (2024c) extend the LLM to perceive spatial audio information obtained from microphone array recordings.

As the visual frame sequence is often paired with audio in real-world video recordings, some studies investigate understanding non-silent video. Vid2Seq (Yang et al., 2023) utilizes the transcription of human speech to enhance video captioning. video-SALMONN (Sun et al., 2024) uses a multi-resolution causal Q-Former to understand audio and video simultaneously. The Google Gemini model achieves video understanding as a native multimodal LLM built upon text, audio, and visual tokens (Team et al., 2024). AVicuna (Tang et al., 2024d) achieves audio-visual temporal understanding by introducing pseudo-untrimmed video data. Video-LLaMA (Zhang et al., 2023) and Video-LLaMA 2 (Cheng et al., 2024) directly concatenating audio and visual tokens for joint audio and video understanding. InternVideo2 (Wang et al., 2024b) aligns video to audio events, speech and text through cross-modal contrastive learning for joint audio-video understanding.

## 2.2 RL FOR LLMs

RL with human feedback (RLHF) (Ouyang et al., 2022) is commonly used to enhance text-based LLMs, with early efforts applying PPO (Schulman et al., 2017) alongside a reward model trained on human preference data. Building on this, DPO (Rafailov et al., 2024) proposes that the LLM itself can serve as a reward model, using paired preference data to optimize the model without the need for an external reward model. KTO (Ethayarajh et al., 2024) further eliminates the need for paired preference data. Expanding on this, RLAIF (Lee et al., 2023) takes a cost-efficient approach by utilizing feedback generated automatically by models, reducing reliance on human involvement.

## 3 METHODS

### 3.1 MODEL ARCHITECTURE

The overall architecture of our model is illustrated in Fig. 1. The paired sequences of audio and visual frames from each video are fed into the audio and visual encoders separately. Users can provide textual prompts to guide the model in performing specific tasks based on the video content. This structure is implemented by incorporating a separate audio encoder branch to a pre-trained visual LLM, which enables the model to process and understand paired audio-visual sequences without degrading its visual performance.

In this structure, audio and visual tokens are computed independently in their respective branches. For the visual branch, the input visual frame sequence is first downsampled at a fixed frame rate of $\phi$ frame/second, and the total number of frames to sample is $n = \phi T$, where $T$ is the duration of the input video in seconds. Let $m$ be the maximum number of frames to sample based on the resource constraint. If $n > m$, the frame rate is further reduced to $\phi' = \lfloor m/T \rfloor$, resulting in $n = \phi'T \leq m$. Let $\mathbf{I}_i$ be the $i$th sampled visual frame, and each visual frame in $\mathbf{I}_1, \mathbf{I}_2, \ldots, \mathbf{I}_n$ is transformed to visual tokens independently using a pre-trained visual encoder $\text{Encoder}_{\text{Visual}}$ followed by a visual modality aligner $\text{Aligner}_{\text{Visual}}$, as shown in Eqn. (1):

$$\mathbf{H}_i^{\text{Visual}} = \text{Aligner}_{\text{Visual}}(\text{Encoder}_{\text{Visual}}(\mathbf{I}_i)), \ \ 1 \leq i \leq n, \tag{1}$$

where $\mathbf{H}_{\text{V}i}$ represents the visual tokens corresponding to $\mathbf{I}_i$.

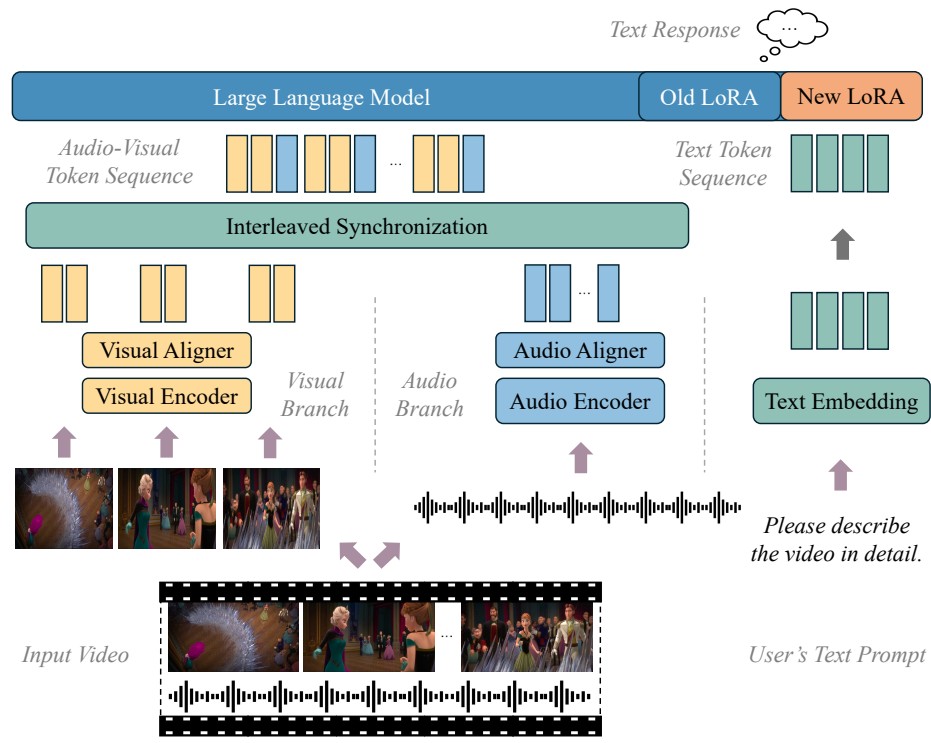

Figure 1: The overall structure of video-SALMONN 2. The input video is processed separately by the visual and audio branches, generating visual and audio tokens from the visual and audio frame sequences. Next, the visual and audio tokens are interleaved synchronously, and combined with the tokens of the text prompt to form the input to the LLM backbone.

The audio frame sequence $\mathbf{S}$ is fed into a pre-trained audio encoder $\text{Encoder}_{\text{Audio}}$. Since $\text{Encoder}_{\text{Audio}}$ may have a maximum processing duration $t_{\max}$, the audio will be sliced into $l = \lceil T/t_{\max} \rceil$ segments of $t_{\max}$-length and processed separately by $\text{Encoder}_{\text{Audio}}$, as shown in Eqn. (2):

$$\mathbf{Z}_j^{\text{Audio}} = \text{Encoder}_{\text{Audio}}(\mathbf{S}_{(j-1) \times t_{\max} : j \times t_{\max}}), \ \ 1 \leq j \leq l, \tag{2}$$

where $\mathbf{Z}_j^{\text{Audio}}$ is the audio feature vector output by the audio encoder of the $j$th audio segment.

As suggested by Yu et al. (2024), a segment-level positional embedding is added before the modality aligner to improve the performance of long-form audio. Denote $\mathbf{Z}_j^{\text{Pos}}$ as the segment-level position embedding matrix corresponding to the position $j$, $\text{Concat}(\cdot)$ as the concatenation operation along the time dimension, and $\text{Aligner}_{\text{Audio}}$ as the audio modality aligner. The audio token sequence $\mathbf{H}^{\text{Audio}}$ for the whole audio can be computed as Eqn. (3)–(5) shown:

$$\tilde{\mathbf{Z}}_j^{\text{Audio}} = \mathbf{Z}_j^{\text{Audio}} + \mathbf{Z}_j^{\text{Pos}}, \ \ 1 \leq j \leq l \tag{3}$$

$$\tilde{\mathbf{Z}}^{\text{Audio}} = \text{Concat}(\tilde{\mathbf{Z}}_1^{\text{Audio}}, \tilde{\mathbf{Z}}_2^{\text{Audio}}, \dots, \tilde{\mathbf{Z}}_l^{\text{Audio}}) \tag{4}$$

$$\mathbf{H}^{\text{Audio}} = \text{Aligner}_{\text{Audio}}(\tilde{\mathbf{Z}}^{\text{Audio}}). \tag{5}$$

Next, the audio and visual tokens are interleaved chronologically to form the input audio-visual token sequence $\mathbf{H}$ fed into the LLM backbone, and $\mathbf{H}$ is obtained based on Eqn. (6)–(8) by

$$\alpha_i = l \cdot i/n, \ \ 1 \leq i \leq n \tag{6}$$

$$\mathbf{H}_i = \text{Concat}(\mathbf{H}_i^{\text{Visual}}, \mathbf{H}_{\alpha_{i-1} : \alpha_i}^{\text{Audio}}), \ \ 1 \leq i \leq n \tag{7}$$

$$\mathbf{H} = \text{Concat}(\mathbf{H}_1, \mathbf{H}_2, \dots, \mathbf{H}_n). \tag{8}$$

Finally, the text-based backbone LLM is required to generate a text response $\hat{\mathbf{Y}}$ given the user's text prompt $\mathbf{P}$ and the audio-visual token sequence $\mathbf{H}$:

$$\hat{\mathbf{Y}} = \arg\max_{\mathbf{Y}} P(\mathbf{Y}|\mathbf{P}, \mathbf{H}). \tag{9}$$

## 3.2 TRAINING STRATEGIES

To introduce audio perceptual capabilities to the visual LLM, we employ a multi-stage training approach that enables the model to fully utilize audio information for video understanding while maintaining its performance in processing visual data. Building on a well-trained visual LLM, the training proceeds through several stages: audio modality alignment, audio-visual SFT, RL based on the proposed mrDPO, and the newly introduced rebirth tuning. The pre-trained visual LLM is designed to understand video frames by integrating pre-trained components such as the LLM, visual aligner, and visual encoder, and both the LLM and the video branch frozen during training to avoid catastrophic forgetting. Similarly, the parameters of the audio encoder remain fixed, as it has been pre-trained on a large-scale audio dataset and possesses strong audio perception capabilities.

Audio modality alignment extends the visual LLM by adding a parallel audio branch, enabling auditory perception capabilities. During this stage, only the audio aligner is trained on a large audio dataset, while the rest of the model remains frozen to preserve its original visual understanding performance. Focusing on learning the audio branch, only audio data is used in this stage. Speech recognition and audio captioning are trained in this stage. The cross-entropy loss is used as the loss function and the reference speech transcriptions and audio captions are used as ground-truth labels.

After audio modality alignment, the backbone LLM can recognize both visual and audio tokens. However, due to the lack of training with paired audio and visual token sequences, the model is not yet capable of synchronizing and integrating audio-visual information for comprehensive video understanding. To address this, we conduct audio-visual SFT using annotated video data. In the SFT stage, the model is trained using cross-entropy loss, with video descriptions serving as ground-truth labels that incorporate both audio and visual information. To enhance the backbone LLM's ability to process audio-visual token sequences, the LLM is jointly trained with LoRA (Hu et al., 2022) during this stage. The audio aligner is trained to map the output of the audio encoder to the input representation space of the LLM, facilitating the backbone LLM's interpretation of audio tokens.

Although the model demonstrates the ability to describe synchronized audio-visual information in video after SFT, several issues persist, including missing information, hallucinations, and repetitive decoding. To address these shortcomings, we apply RL based on mrDPO to improve the model's performance. Additionally, we introduce rebirth tuning after RL to further enhance the model's performance in non-captioning tasks. Fig. 2 provides an overview of the entire training process involving mrDPO and rebirth tuning, with further details explained in Section 3.3 and 3.4.

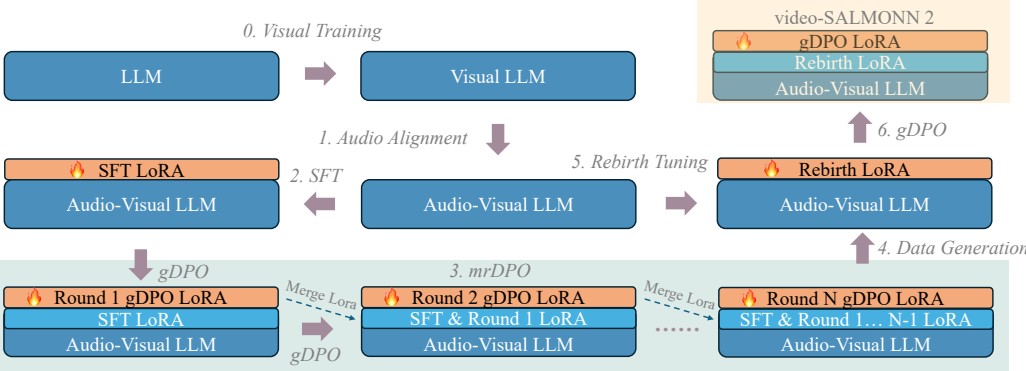

Figure 2: An overview of the training process including audio modality alignment, SFT, mrDPO, and rebirth tuning. LoRA is introduced during the SFT stage. In each round of the mrDPO training, a new LoRA proxy is added while the old LoRA is merged into the LLM. After mrDPO, the model generates labels for new training data for rebirth tuning based on the initial audio-visual LLM. The final video-SALMONN 2 model is obtained by another gDPO training after rebirth tuning.

## 3.3 MULTI-ROUND DPO

We aim to leverage DPO to improve the quality of video captions generated by the model. To establish an effective method for evaluating the completeness of video captions, we propose using

atomic events as a bridge to automatically assess the preference of caption samples through artificial intelligence (AI) feedback, guiding the model to produce more accurate and detailed video captions.

Before DPO training, ground-truth video captions are first decomposed into atomic events using a powerful text LLM, such as GPT-4, to provide references for evaluating the captions generated by the model. Next, we apply the nucleus sampling method (Holtzman et al., 2020) to generate captions from the model's output distribution for the input video. Depending on the input prompt, the captions can be either *global*, summarizing the entire video, or *local*, focusing on specific time intervals within the video. For each video, two *global* captions and two *local* captions are sampled.

For both *global* and *local* caption pairs, the atomic events from the ground-truth video captions are used to identify the preferred sample within each pair. Specifically, we input a video caption and the corresponding list of ground-truth atomic events into a powerful text LLM, such as GPT-4. The LLM is prompted to identify which atomic events from the ground-truth list are missing in the caption and which events described in the caption are absent from the ground-truth list. These are referred to as "missing events" and "hallucination events", respectively. The missing and hallucination rates for each caption are calculated by dividing the number of each type of event by the total number of atomic events in the ground-truth caption. The total error rate of a caption is then obtained by summing its missing and hallucination rates. Additionally, we penalize redundancy by measuring a *repetition rate* for each generated caption, as detailed in Appendix B. The caption with a relatively lower total error rate and repetition rate within a pair is selected as the preferred sample for DPO. To improve efficiency and mitigate the impact of LLM evaluation noise, caption pairs with small differences in these metrics are excluded from the DPO training set.

Unlike previous approaches that applied only single-round DPO to multimodal LLMs, we introduce a multi-round strategy, as prolonged offline training with a single round fails to optimize the model effectively due to the reference model being biased against the most recent model update in the DPO algorithm. In the multi-round framework, at each $t$th round, the following steps are taken to perform DPO training for the current round.

1. First, pre-trained LoRA module $\Delta_{t-1}$ is merged into the LLM backbone $\Lambda_{t-1}$ to derive a new LLM backbone $\Lambda_t$ that is equivalent to $\Lambda_{t-1}$ with $\Delta_{t-1}$, based on Eqn. (10):

$$\mathbf{W}_t = \mathbf{W}_{t-1} + \alpha \mathbf{A}_{t-1} \mathbf{B}_{t-1}, \tag{10}$$

   where $\mathbf{W}_t$ and $\mathbf{W}_{t-1}$ are the weight parameters to adapt in $\Lambda_t$ and $\Lambda_{t-1}$, $\alpha$ is the scaling factor of LoRA, $r$ is the rank of LoRA, $d$ is the dimension of $\mathbf{W}_{t-1}$. $\mathbf{A}_{t-1} \in \mathcal{R}^{d \times r}$ and $\mathbf{B}_{t-1} \in \mathcal{R}^{r \times d}$ are the low-rank matrix parameters of LoRA in the previous round $t-1$, and $\mathbf{W} \in \mathcal{R}^{d \times d}$ is the parameter of LLM backbone.

2. Next, a new randomly initialized LoRA module $\tilde{\Delta}_t$ is added to the LLM backbone, forming the new policy model $\Lambda_t$ for round $t$. Only LoRA parameters $\tilde{\Delta}_t$, referred to as the LoRA proxy, are updated for round $t$ with other LoRA parameters fixed. To address the issue of the increasing difference between the reference and policy models caused by freezing the reference model in standard DPO, $\Lambda_t$ is used as the updated reference model used in round $t$.

3. At last, $\tilde{\Delta}_t$ is trained to obtain a well-trained $\Delta_t$, which can be achieved using the standard DPO loss. However, after multiple training rounds, the model starts to produce unnatural language patterns such as unintelligible or meaningless sentences. To alleviate this issue by stabilizing the training, a guided DPO (gDPO) loss is proposed as

$$\mathcal{L}_{\mathrm{gDPO}}(\pi_\theta; \pi_{\mathrm{ref}}) = -\mathbb{E}_{(\mathbf{x}, \mathbf{y}_{\mathrm{win}}, \mathbf{y}_{\mathrm{lose}}) \sim \mathcal{D}} \left[ \log \sigma \left( \beta \log \frac{\pi_\theta(\mathbf{y}_{\mathrm{win}} \mid \mathbf{x})}{\pi_{\mathrm{ref}}(\mathbf{y}_{\mathrm{win}} \mid \mathbf{x})} - \beta \log \frac{\pi_\theta(\mathbf{y}_{\mathrm{lose}} \mid \mathbf{x})}{\pi_{\mathrm{ref}}(\mathbf{y}_{\mathrm{lose}} \mid \mathbf{x})} \right) \right]$$
$$+ \lambda \, \mathbb{E}_{(\mathbf{x}, \mathbf{y}_{\mathrm{gt}}) \sim \mathcal{D}_{\mathrm{gt}}} \log \pi_\theta(\mathbf{y}_{\mathrm{gt}} | \mathbf{x}), \tag{11}$$

   where $\pi_\theta = \{\Lambda_t, \tilde{\Delta}_t\}$ and $\pi_{\mathrm{ref}} = \Lambda_t$ represent the policy and reference models for round $t$ respectively. Variables in the first term follow the definitions in the standard DPO loss (Rafailov et al., 2024). In the second term corresponding to cross-entropy learning towards the ground-truth video captions, $\lambda$ is the weight of the second term in the overall loss, $\mathcal{D}_{\mathrm{gt}}$ denotes the SFT training dataset, and $(\mathbf{x}, \mathbf{y}_{\mathrm{gt}})$ corresponds to a video and its paired ground-truth text description. Each mini-batch of training samples is randomly selected from $\mathcal{D}_{\mathrm{gt}}$.

These steps complete the training for a single round. Our proposed mrDPO is implemented by repeating these steps across multiple rounds. Notably, by merging $\Delta_{t-1}$ into $\Lambda_{t-1}$ and equipping

the resulting $\Lambda_t$ with a new LoRA module $\tilde{\Delta}t$, the new $\tilde{\Delta}t$ functions as a LoRA proxy for parameter updates. This proxy helps regularize the training by introducing a new random initialization at each round of mrDPO.

### 3.4 REBIRTH TUNING

After mrDPO training, the model exhibits significant improvements in captioning abilities, highlighting the effectiveness of mrDPO in enhancing audio-visual understanding. While achieving strong performance as measured by low event missing and hallucination rates, the model begins to produce unnatural or unexpected text patterns after multiple DPO rounds. These include phrases like "[VideoDescription]: ", "###Audio Description", and sequences of uninterpretable symbols. This behaviour may stem from discrepancies between LLM-generated feedback and actual human feedback for such unnatural text, leading the model to inadvertently adopt these patterns as training progresses.

This paper proposes Rebirth Tuning, a method designed to address the degradation of language generation abilities in mrDPO while preserving its benefits in ensuring information completeness and correctness. Rebirth tuning employs teacher-forcing training on self-generated data, promoting a more stable learning process. Teacher-forcing, which guides the model to predict the next token, helps prevent it from converging on limited and repetitive patterns. More specifically, before applying rebirth tuning, mrDPO is stopped once we observe a significant decline in the model's language capabilities. The final iteration of the model, which excels at generating complete and accurate video descriptions, is then used to label a large dataset of videos. Since the model's language abilities remain relatively intact, natural and fluent descriptions can be easily filtered by detecting problematic patterns, with the remaining high-quality descriptions used as training data for rebirth tuning.

Rebirth tuning is applied to the model after audio modality alignment, allowing it to be "reborn" from self-generated high-quality data to enhance video understanding. Following rebirth tuning, the model not only avoids catastrophic forgetting of non-captioning abilities but also supports the development of the next generation of models by applying mrDPO, followed by the subsequent stage of rebirth tuning.

## 4 EXPERIMENTAL SETUP

### 4.1 MODEL SPECIFICATIONS

video-SALMONN 2 is built on an internally trained high-performance visual LLM. It uses SigLIP (Zhai et al., 2023) as the visual encoder, Qwen 1.5 with 7B parameters as the backbone LLM, and two linear layers with GELU activation function (Hendrycks & Gimpel, 2016) as the visual aligner. The model processes video frames at a per-second frame rate of 1 (*i.e.*, $\phi = 1$), and can handle up to 30 frames. For videos longer than 30 seconds, 30 frames are uniformly sampled from the video.

For the audio branch, we use the Whisper-Large-v3 encoder (Radford et al., 2023) as the audio encoder, and a window-level Q-Former (Tang et al., 2024a) with a window length of 0.2 seconds as the audio aligner, producing a total of 150 audio tokens for a 30-second input. The Whisper encoder has a maximum processing duration of $t_{\max} = 30$ seconds. The rank $r$ and scaling factor $\alpha$ of LoRA are set to 256 and 2.0, respectively. During training, the visual encoder, visual aligner, audio encoder, and LLM remain frozen.

### 4.2 DATA AND TRAINING SPECIFICATIONS

During the audio modality alignment stage, LibriSpeech-960h (Panayotov et al., 2015) and Audio-Caps (Kim et al., 2019) are used to train the audio aligner. LibriSpeech-960h is utilized for speech recognition training, while AudioCaps is employed for audio captioning training.

In the audio-visual SFT stage, experiments are conducted using an internal video dataset that will be released upon acceptance. A total of 13k videos with rich audio information are automatically selected, and high-quality audio-visual captions are regenerated with the assistance of GPT-4o (OpenAI et al., 2024), Whisper-Large-v3 (Radford et al., 2023), and SALMONN (Tang et al., 2024b). The detailed pipeline is described in Appendix C. Additionally, to further enhance the quality of the SFT data, around 1.5k video captions were manually refined.

In the mrDPO stage, two kinds of tasks are studied: global captioning for the whole video and local captioning for a given time interval. Before each round, a pair of captions for both global and local captioning are sampled from the model for each video in SFT data, respectively. We consider the information missing rate, hallucination rate, and repetition rate to determine whether a sample pair is suitable for DPO and determine the chosen and rejected samples if yes. The selecting methods for each round are listed in Appendix F. We maintain a learning rate of $5 \times 10^{-6}$ and set the weight $\lambda$ in Eqn.(11) to 0.1 throughout the entire mrDPO stage. Additionally, we provide the training curves in Appendix G to support reproducibility.

After mrDPO, the language abilities of the model are reduced. We stop DPO training once significant degradation in language abilities is detected. The checkpoint of the last DPO round is used to label captions of a large number of videos. Unnatural captions are eliminated, and the remaining high-quality captions form the training data for the rebirth tuning stage.

For the test dataset, we curated a video captioning benchmark to evaluate the event missing rate (Miss), hallucination rate (Hall), and text repetition rate (Rep). Details of the test data and evaluation process can be found in Appendix D and Appendix E, respectively. The benchmark consists of 483 carefully selected videos, each labelled with complete audio-visual captions by human annotators. Atomic events for the test dataset were initially obtained using GPT-4o and then manually refined. For local captioning, we used Gemini-1.5-Pro (Team et al., 2024) to tag the start and end times of each event within specific time intervals. Since Gemini could not process some videos, only 457 videos were used for the local captioning evaluation.

Regarding training settings, we conducted audio modality alignment using 8×A100 GPUs for 35k steps and audio-visual SFT using 16×A100 GPUs for 4 epochs. Each mrDPO round was trained with 64×A100 GPUs for 1k steps. After six rounds of mrDPO training, rebirth tuning was performed. During the rebirth tuning stage, we used 64×A100 GPUs and trained for 4 epochs. The batch size per device was set to 1, making the total batch size equal to the number of devices. The final video-SALMONN 2 model was derived by applying a single round of gDPO training to the model obtained after rebirth tuning.

## 5 EXPERIMENTAL RESULTS

### 5.1 OVERALL RESULTS

The results of our video captioning benchmarks are presented in Table 1. video-SALMONN 2 outperforms other models in both information missing and hallucination rates for global and local captioning. Among existing open-source multimodal LLMs, few can provide detailed and accurate video descriptions, whether purely visual models like Video-LLaVA (Lin et al., 2024) and VILA (Lin et al., 2023), or audio-visual models like Video-LLaMA 2 (Cheng et al., 2024) and video-SALMONN (Sun et al., 2024). Notably, many open-source models, such as Video-LLaVA and Video-LLaMA 2, tend to generate shorter captions, leading to high information missing rates but low hallucination rates. GPT-4o and Gemini-1.5-Pro can generate more detailed captions and are of higher quality than current open-source models. However, the purely visual version of GPT-4o lacks audio comprehension, and Gemini's understanding of visual content is somewhat limited, resulting in both models exhibiting some degree of information missing and hallucination.

Our visual base model, trained on a large dataset of images and silent videos, is capable of generating detailed text descriptions based solely on visual information, with a relatively low information missing rate. However, generating longer texts leads to a higher hallucination rate. After audio modality alignment and audio-visual SFT, the model can leverage audio content to reduce both information loss and hallucinations in its descriptions. However, the inclusion of audio tokens may confuse the visual LLM, resulting in frequent repetition during decoding. Building on the SFT model, we applied mrDPT and rebirth tuning, achieving approximately a 35% absolute reduction in the repetition rate and absolute reductions of around 40% and 20% in the total error rate for global and local captioning, respectively. The final video-SALMONN 2 model outperforms some commercial models like GPT-4o and Gemini-1.5-Pro in video captioning. Inspired by Chai et al. (2024), we also perform Elo ranking with human evaluators, which validates the alignment of our proposed metrics with human perception and highlights the strong video captioning capabilities of video-SALMONN 2. Further details are provided in Appendix H. In addition, as an audio-visual LLM, video-SALMONN 2 retains strong visual understanding capabilities and performs well on various video benchmarks, such as Video-MME (Fu et al., 2024a). Appendix I shows more details.

Table 1: Results of our benchmark for detailed video captioning evaluation. "A" and "V" refer to the audio and visual modalities respectively. The repetition rate (Rep), event missing rate (Miss), hallucination rate (Hall), and total error rate (Total = Miss + Hall) are assessed for the captions. video-SALMONN 2, which undergoes an additional round of gDPO after rebirth tuning, achieves the best performance in both global and local captioning, with the lowest total error rates.

| Model | Modality | Global | | | | Local | | |
|---|---|---|---|---|---|---|---|---|
| | | %Rep↓ | %Miss↓ | %Hall↓ | %Total↓ | %Miss↓ | %Hall↓ | %Total↓ |
| GPT-4o Visual | V | 3.6 | 16.6 | 17.2 | 33.8 | 35.3 | 30.7 | 66.0 |
| Gemini-1.5-Pro | A + V | 1.3 | 21.8 | 16.5 | 38.3 | 36.9 | 17.2 | 54.1 |
| 7B Video-LLaVA | V | 13.2 | 65.3 | **5.4** | 70.7 | 59.1 | **9.4** | 68.5 |
| 8B VILA | V | 4.5 | 39.3 | 18.6 | 57.9 | 47.9 | 23.4 | 71.2 |
| 7B Video-LLaMA 2 | A + V | 5.7 | 56.8 | 8.9 | 65.7 | 47.6 | 14.3 | 61.9 |
| 13B video-SALMONN | A + V | **1.2** | 52.1 | 26.6 | 78.7 | 47.8 | 40.7 | 88.4 |
| 7B Ours-Visual Base | V | 11.8 | 29.8 | 30.0 | 59.7 | 36.1 | 46.1 | 82.2 |
| 7B Ours-SFT | A + V | 36.0 | 26.7 | 26.9 | 53.6 | 30.8 | 33.3 | 64.0 |
| 7B video-SALMONN 2 | A + V | 1.4 | **6.9** | 6.8 | **13.7** | **22.2** | 21.4 | **43.6** |

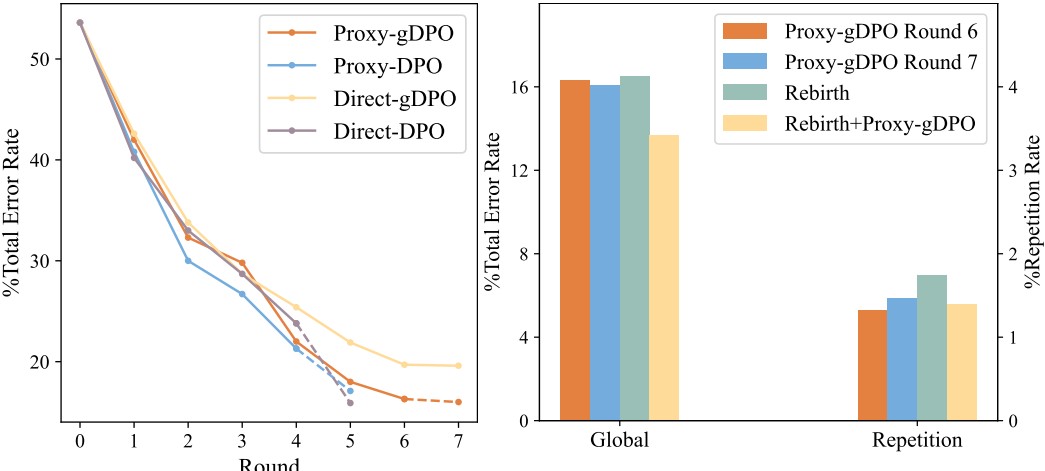

(a) Comparison of different mrDPO settings.  (b) gDPO with LoRA proxy vs. Rebirth tuning.

Figure 3: The comparison of different methods is shown in Fig. 3a. All models are trained either until convergence or until they begin to frequently generate unnatural patterns. The dashed line indicates that this round of DPO resulted in a high frequency of unnatural text generation and therefore, cannot be considered a valid performance indicator for the model. Fig. 3b demonstrates that DPO after rebirth tuning achieves better performance, while mrDPO provides only minimal gains.

## 5.2 ANALYSIS OF MRDPO

This section investigates various training strategies in mrDPO. For loss functions, we compared the standard DPO loss with our proposed gDPO loss, which incorporates an additional regularization term based on the ground-truth captions. Moreover, we evaluated the LoRA proxy against direct tuning of the model's original LoRA, referred to as "Proxy" and "Direct," respectively. Figure 3a illustrates the total error rates for global video captioning across different methods over multiple DPO rounds. The training was halted when unnatural captions began to appear frequently. Examples of these unnatural cases are provided in Appendix L.

First, mrDPO training consistently outperformed single-round DPO training across various settings. While single-round DPO effectively reduces the total error rate, the increasing divergence between the most recent policy model and the reference model indicates that the assumption of their approximate similarity no longer holds. As a result, model performance cannot be continuously improved through a single round of DPO training due to this growing deviation.

Besides, in terms of the loss functions, the classical DPO loss shows the fastest improvement in captioning metrics, but unfortunately, it also quickly leads to outputs with frequent unnatural patterns.

This is likely because the model only sees self-generated labels rather than ground-truth labels. By incorporating loss on ground-truth labels, gDPO makes the training process more stable, allowing the model to generate text responses without unnatural patterns for a longer period of training across multiple DPO rounds. This stability also preserves the model's potential for further improvement, with a significant drop in the error rate observed after three rounds of mrDPO.

Using LoRA proxies that randomly initialize a new LoRA in each DPO round, is found to be more beneficial for mrDPO performance compared to directly training the same LoRA, especially after over 4 gDPO rounds. This demonstrates that the regularization effect introduced by the LoRA proxies helps prevent the model from converging to a local optimum, thereby mitigating overfitting. Since gDPO with LoRA proxy performs the best for mrDPO, we use the model after six gDPO rounds using LoRA proxy to generate captions for a large number of videos. After excluding unnatural patterns, a total of 180k video captions remain for rebirth tuning.

We also provide some video captioning examples of models at different training stages to intuitively and qualitatively see the improvement brought by mrDPO. Refer to Appendix K.

### 5.3 ANALYSIS OF REBIRTH TUNING

Table 2: The appearance rate of unnatural caption for global. We detect specific patterns that are viewed as unnatural and count the frequency of occurrence of specific patterns on the test set over mrDPO using gDPO with LoRA proxy. The results prove that mrDPO leads to a significant increase in the appearance of unnatural captions, especially in the final converging rounds.

| Stage | SFT | #Rounds of mrDPO | | | | | | | Rebirth Tuning |
|---|---|---|---|---|---|---|---|---|---|
| | | 1 | 2 | 3 | 4 | 5 | 6 | 7 | |
| %Unnatural Rate↓ | 0.0 | 0.0 | 0.0 | 1.9 | 0.9 | 2.9 | 2.1 | 12.0 | 0.0 |

While multiple rounds of mrDPO with LoRA proxy significantly improve video captioning performance, they also lead to an increasing frequency of unnatural patterns in text responses. Table 2 shows the occurrence rate of these unnatural patterns in global captioning after each training stage. Through rebirth tuning, the backbone LLM discards the LoRA proxies and restores its ability to generate fluent captions. Additionally, the careful selection of rebirth-tuning data enhances data quality, ensuring the model is fine-tuned with superior data, further boosting its overall performance. Rebirth tuning also partially restores the model's performance on video QA benchmarks. Detailed results are provided in Appendix J.

Another notable effect of rebirth tuning is to sustain continued training. As shown in Fig. 3a, in later rounds of mrDPO, the model starts to gain less in each round and eventually converges. The decline in the ability to generate fluent text responses is also more likely to occur in these later rounds, suggesting that the model has fallen into a local minimum after mrDPO. However, after the rebirth tuning stage, where only teacher-forcing training is applied, the model escapes the local optimum from previous training and becomes receptive to further optimization with DPO training. Fig. 3b compares gDPO after rebirth tuning with six rounds of gDPO with LoRA proxy. It is observed that only minimal improvement can be achieved after sufficient gDPO rounds in terms of the total error rate for global captioning, while an extra gDPO stage following rebirth tuning yields significant performance gains once again. This suggests the potential of iterating mrDPO and rebirth tuning.

## 6 CONCLUSIONS

This work introduces video-SALMONN 2, a powerful audio-visual LLM designed for detailed video captioning, and proposes the mrDPO method. To our knowledge, this is the first study of applying RL to audio-visual LLMs in literature. New metrics are designed to evaluate the information missing and hallucination rates in video captions, which are used to guide sample selection for DPO. To further stabilize training, the setting with novel gDPO and LoRA proxy is introduced. After mrDPO, we propose a novel rebirth tuning method to restore LLM's performance on non-captioning tasks. As a result, video-SALMONN 2 demonstrates significant improvements in video captioning, outperforming notable models such as GPT-4o and Gemini-1.5-Pro, and setting a promising direction for achieving detailed and accurate video captioning for video understanding.

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

## A    PIPELINE FOR GETTING ATOMIC EVENTS IN A TIME INTERVAL

Gemini-1.5-Pro is used to obtain the atomic events in some time intervals. We input the video and all its atomic events labelled by GPT to Gemini and ask it to tag the beginning and start time of each atomic event. Events are then selected if their time intervals overlap with the given time interval. We have checked this process to confirm that the atomic events obtained for the given time interval are roughly accurate.

## B    CALCULATION PROCEDURE OF THE TEXT REPETITION RATE

The procedure to calculate the repetition rate of a long and detailed text is shown as follows:

1. Split the text into short phrases by punctuation;

2. Counting the number of occurrences of each phrase;

3. The number of recurring phrase words divided by the total number of words in the text is the repetition rate.

## C    PIPELINE FOR LABELLING HIGH-QUALITY AUDIO-VISUAL CAPTIONS

To curate training data for supervised fine-tuning, we employ GPT-4o to label the visual content in each frame, while SALMONN-13B and Whisper-Large-v3 are used to annotate the speech content and audio events in the audio track. This process is illustrated in Figure 4. Our initial aim is to automatically filter out videos that contain limited speech. We begin by slicing each video into 10-second segments, with the audio from each segment analyzed by SALMONN to generate automatic audio captions (AAC). These captions help us filter out videos that lack descriptive speech such as "A man is speaking" or "A woman says...". This initial filtering step is somewhat coarse.

Next, the audio from each segment is processed by Whisper to produce automatic speech recognition (ASR) results. If the transcribed text is too brief or nonsensical, the corresponding video is deemed to lack rich audio content and is excluded from further consideration. For a video to be labelled, all of its segments should pass this exclusion criterion.

The segments from the remaining videos are then sampled at a rate of 1 fps and fed into GPT-4o to extract segment-level visual captions. Ultimately, the segment-level visual captions, AAC results, and ASR results form the input to GPT-4o concurrently to generate a detailed global audio-visual caption for each video.

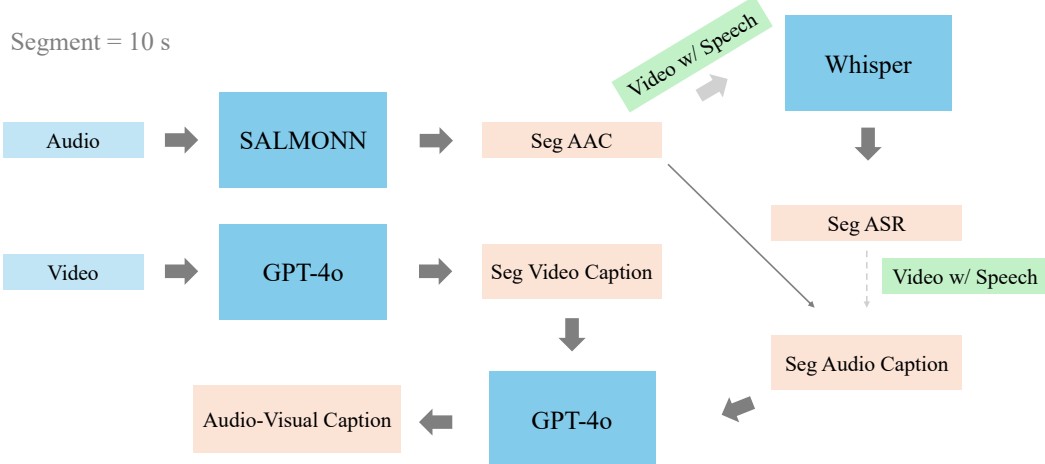

Figure 4: The pipeline for labelling videos with high-quality audio-visual captions.

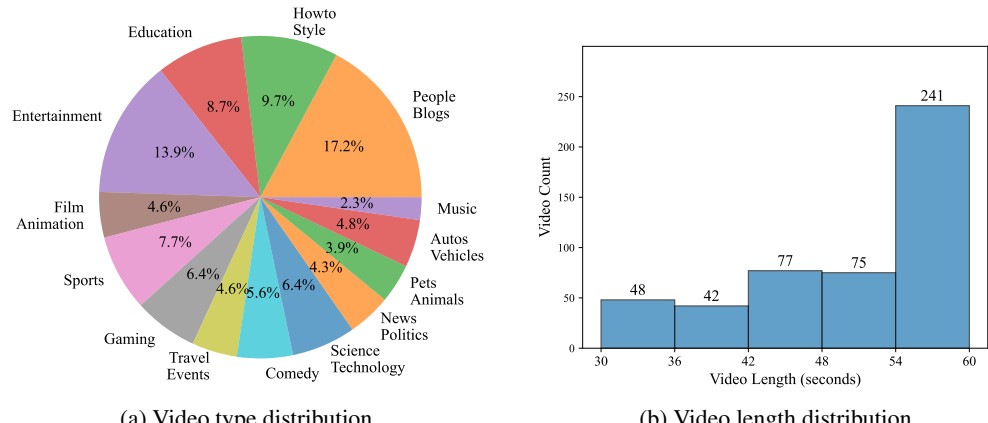

(a) Video type distribution.  (b) Video length distribution.

Figure 5: The basic information of our benchmark data.

## D    ABOUT THE TEST DATASET

Figure 5 shows the video type distribution and the video length distribution of our caption benchmark. The benchmark covers 14 different fields. All the videos are between 30s to 60s, with an average duration of 51s. Table 3 shows more statistics about the labelled captions and atomic events of the benchmark. Specifically, there are 6.1 audio-related atomic events per video, including 4.6 speech-related events and 1.5 non-speech events.

Table 3: Statistics about the labelled captions and atomic events of the benchmark.

| #Sample | #Vocabulary | #Word | Average Statistics | |
| --- | --- | --- | --- | --- |
| | | | Caption Length | #Atomic Event |
| 483 | 17137 | 296,938 | 615 words | 34.2 |

## E    PROCESS OF EVALUATING DETAILED CAPTIONS

To evaluate the specific video caption generated by our model, we first use GPT-3.5 or GPT-4o to split the labelled caption of this video into several atomic events, where we use GPT-4o for the test set and GPT-3.5 for the RL training set. Then, the list of atomic events and the caption to be evaluated are simultaneously fed into GPT-3.5 to determine what events in the atomic event list are missed and what events in the caption are hallucinated. Specifically, we ask GPT-3.5 to list all the missing events and hallucination events for better evaluation precision. Note that events that are described incorrectly are also regarded as hallucinations. The quotient between the number of missing or hallucination events and the number of all events in the video is the final missing or hallucination rate. For more robust testing, GPT-3.5 is used to evaluate 7 times for each caption and the medium number of the metric is reported. We have manually confirmed that the score GPT-3.5 gives is roughly plausible.

## F    SAMPLES SELECTING METHODS FOR DPO OF EACH ROUND

To achieve better performance and training efficiency, we take a specially designed strategy to select proper preference pairs. A sample pair is selected if one sample is better than the other in all metrics with a threshold. For global captioning, we consider global error rate $\Delta e_g$ and global repetition rate $\Delta r_g$, while for local captioning we consider local error rate $\Delta e_t$ and local repetition rate $\Delta r_t$. Table 4 shows the threshold used in each round.

Table 4: The data selecting threshold used in each DPO round. A negative number means that the chosen sample can be worse than the rejected sample in this metric to some degree.

| DPO Round | Threshold Used | | | |
|---|---|---|---|---|
| | $\Delta e_g$ | $\Delta r_g$ | $\Delta e_t$ | $\Delta r_t$ |
| 1 | $\geq 5\%$ | $\geq 1\%$ | $\geq 20\%$ | $\geq 1\%$ |
| 2 | $\geq 20\%$ | $\geq -1\%$ | $\geq 45\%$ | $\geq 0$ |
| 3 | $\geq 20\%$ | $\geq -1\%$ | $\geq 45\%$ | $\geq 0$ |
| 4 | $\geq 20\%$ | $\geq -1\%$ | $\geq 45\%$ | $\geq 0$ |
| 5 | $\geq 20\%$ | $\geq -1\%$ | $\geq 45\%$ | $\geq 0$ |
| 6 | $\geq 25\%$ | $\geq -1\%$ | $\geq 45\%$ | $\geq 0$ |
| 7 | $\geq 30\%$ | $\geq -1\%$ | $\geq 45\%$ | $\geq 0$ |

## G  TRAINING CURVES IN MRDPO

In mrDPO, we keep the learning rate to 5e-6, and set the weight $\lambda$ in Eqn. 11 to 0.1. The training curves of the loss and rewards in each DPO round are shown in Figure 6.

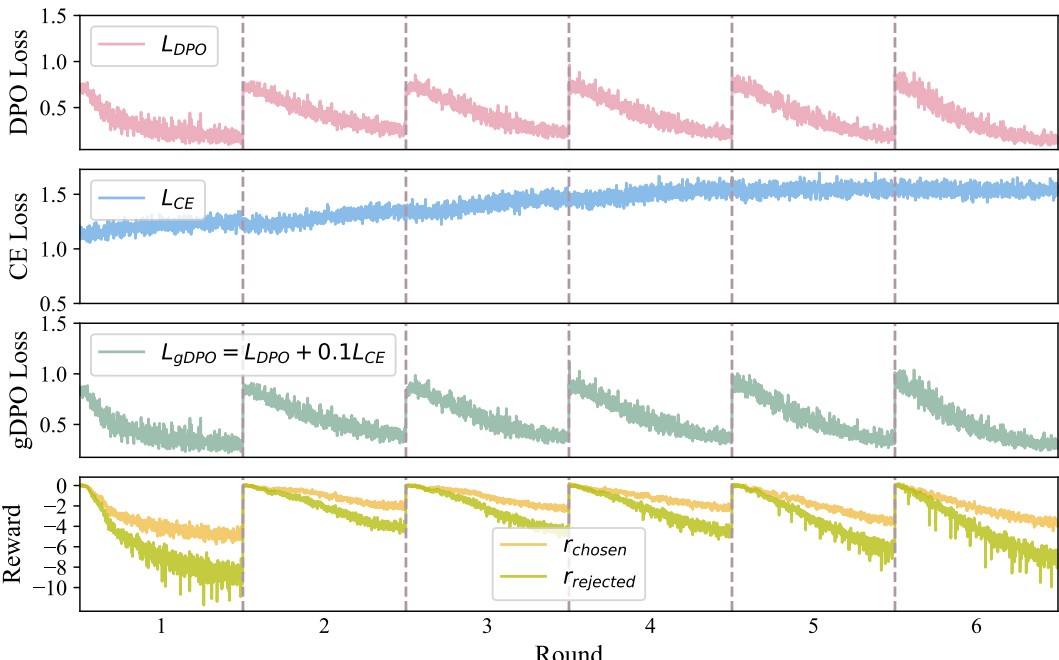

Figure 6: Training curves of the loss and rewards in each DPO round. We use gDPO loss for training, which is equal to sum of the classical DPO loss and the cross entropy loss with the ground-truth captions, as Equ 11 shows.

## H  ELO RANKING RESULTS

We perform the Elo ranking by human to rate the video captioning capability of Gemini-1.5-pro, GPT-4o, our visual base model, and the final video-SALMONN 2. Parameters of the Elo ranking system are provided in Table 5.

We collected 360 pairs of captions generated by different models as simulated matches, where each model generates 180 video captions. We repeated the collected comparison 8 times to form 2880 matches so that stable ranking results can be obtained for our models. Annotators are provided with the video and two corresponding video captions generated by two models respectively, and they are

Table 5: Parameters of the Elo ranking system.

| Parameter | Value |
|---|---|
| Initial ELO mean | 1000 |
| Base of logarithm | 10 |
| Scaling factor | 400 |
| K-factor | 2 |

expected to select the better caption according to the completeness and correctness of the caption. The final Elo rating results and the global caption error rates of each model are provided in Table 6.

Table 6: Elo rating results compared with the global captioning error rates of different models. "Total" represents the total error rate of global video captioning.

| Model | Total%↓ | Elo Rating↑ |
|---|---|---|
| GPT-4o Visual | 33.8 | 1048 |
| Gemini-1.5-pro | 38.3 | 1010 |
| Ours-Visual Base | 59.7 | 888 |
| video-SALMONN 2 | **13.7** | **1054** |

# I  RESULTS ON OTHER VIDEO BENCHMARKS

We also test video-SALMONN 2 on some popular video QA benchmarks. Since QA data is not seen during the mrDPO process, the model's QA abilities decrease a lot after mrDPO. Thanks to the rebirth tuning on captioning and QA, the non-captioning abilities can recover. After one round of gDPO with a LoRA proxy, video-SALMONN 2 finally achieves detailed and accurate captioning while getting competitive results compared to SOTA models of similar size. Video-MME, NeXT-QA, MVBench and VideoVista are test here. Since video-SALMONN 2 cannot support very long audio due to the memory limit, we only use video frames and discard the audio track when testing Video-MME Medium and Long sets. For VideoVista, we only input the model with the video frames as well for too-long videos. Table 7 shows the results.

Table 7: Results of different models on various video QA benchmarks.

| Model (#Params) | Video-MME Acc%↑ | NeXT-QA Acc%↑ | MVBench Acc%↑ | VideoVista Acc%↑ |
|---|---|---|---|---|
| Video-LLaVA (7B) (Lin et al., 2024) | 39.9 | - | - | 53.8 |
| Long-LLaVA (9B) (Wang et al., 2024a) | 52.4 | - | 54.6 | - |
| VideoChat 2 Mistral (7B) (Li et al., 2024a) | 42.3 | **78.6** | **61.3** | 54.9 |
| video-SALMONN 2 (7B) | **54.1** | 71.4 | 51.4 | **65.3** |

In addition, as an audio-visual LLM, being able to understand synchronized audio-visual elements is a unique ability compared with the pure visual LLM. We test our model using an internal benchmark, which is an audio-visual QA test set focusing on audio-visual synchronization. There are mainly two types of questions in this benchmark:

1. When the speaker mentions some specific things, what happens in the video?
2. When some specific things happen in the video, what does the character say?

The accuracy of our model is shown in Table 8. Gemini-1.5-Pro's result is also shown as a comparison:

Although our model's performance on the Synchronized Audio-Visual QA benchmark is slightly below that of Gemini-1.5-Pro, its accuracy remains competitive given its smaller model size and

Table 8: Results of video-SALMONN 2 on our internal Synchronized Audio-Visual QA benchmark.

| Model | Acc%↑ |
|---|---|
| Gemini-1.5-Pro | 88.9 |
| video-SALMONN 2 | 65.8 |

significantly reduced training data. It is worth noting that, due to the limited general speech and audio understanding capabilities of many public video LLMs, there are currently no other models available for evaluation on this internal benchmark.

For a detailed evaluation of video captioning, Chai et al. (2024) introduced the Video Detailed Captions (VDC) benchmark, which uses a large number of question-answer pairs to prompt GPT-4 to assess the quality of video captions. We evaluated Video-SALMONN 2 on the VDC benchmark, and the results, presented in Table 9, demonstrate that our model achieves a competitive score on this challenging benchmark.

Table 9: Results of video-SALMONN 2 on the VDC benchmark. The results of the SOTA Aurora-Cap model on the benchmark is listed as well.

| Model | Camera Acc / Score | Short Acc / Score | Background Acc / Score | Main Object Acc / Score | Detailed Acc / Score |
|---|---|---|---|---|---|
| AuroraCap (Chai et al., 2024) | **43.50 / 2.27** | **32.07 / 1.68** | 35.92 / 1.84 | 39.02 / 1.97 | **41.30 / 2.15** |
| video-SALMONN 2 | 35.81 / 1.94 | 28.77 / 1.52 | **40.83 / 2.13** | **40.67 / 2.12** | 36.67 / 1.96 |

## J  QA PERFORMANCE AT EACH TRAINING STAGE

We test the models on two video question-answering benchmarks: NExT-QA and Video-MME. Since our model is trained on paired audio videos of around 1 minute, we only test the models on the "Short" set of Video-MME to reflect changes in the model's QA capabilities more accurately.

Table 10: Performance of models after different training stages on NExT-QA and Video-MME Short benchmarks.

| Model | NExT-QA ACC%↑ | Video-MME Short ACC%↑ |
|---|---|---|
| Ours-Visual Base | 72.8 | 67.2 |
| Ours-gDPO Round 3 | 72.0 | 67.3 |
| Ours-gDPO Round 6 | 70.2 | 65.3 |
| Ours-Rebirth Tuning | 71.1 | 67.6 |
| Ours-Rebirth Tuning + 1 Round gDPO | 71.4 | 67.0 |

In the first few rounds of mrDPO, the model's QA capability remained largely unchanged. However, after 6 rounds of gDPO, performance on the QA benchmarks dropped, although the model's caption became more complete. After rebirth tuning, the model regained performance on the QA benchmarks.

## K  VIDEO CAPTIONING CASES OF MODELS AT DIFFERENT TRAINING STAGES

Here we show some video captioning cases of models at different training stages. We randomly select a video and prompt our models to caption the video. The atomic events of this video are shown in Figure 7, and the caption results of models at different training stages are shwon in Figure 8, 9, 10 and 11. This is a video sample that we randomly select without cherry picking.

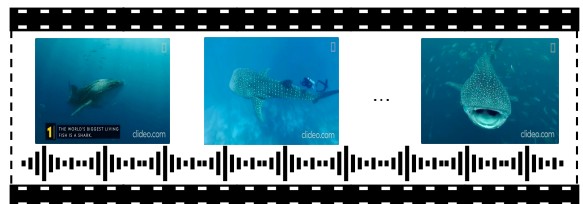

**Atomic Event List**

1. The video begins with a whale shark swimming in the ocean.
2. The whale shark moves slowly through the water.
3. The number '1' in a yellow box apeears in the video.
4. The text 'THE WORLD'S BIGGEST LIVING FISH IS A SHARK.' appears in the video.
5. A vibrant coral reef teeming with small fish appears.
6. A woman says, "The world's biggest living fish is a shark. Of the estimated 34,000 species of fish, the largest are whale sharks."
7. Background music can be heard.
8. A diver swims near the whale shark.
9. The scene changes again, showing a diver swimming near the whale shark.
10. A woman's voice says, "These gentle giants usually grow to about 40 feet long and weigh an estimated 15 tons. Their mouths alone can span 4 feet wide."
11. The background music continues.
12. Text 'clideo.com' can be seen in the bottom right corner.
13. The scene changes, showing a close-up view of the whale shark's mouth.

Figure 7: An example of the atomic event list of a video.

## L  CASES OF UNNATURAL VIDEO CAPTIONS

Unnatural responses might be generated after multiple DPO rounds. Figure 12 shows a very unnatural caption generated by the model after too many rounds of DPO training.

**Output by the model before mrDPO:**

The video begins with a majestic whale shark swimming gracefully in the deep blue ocean. The whale shark, characterized by its distinctive white spots and immense size, moves slowly through the water. The text 'THE WORLD'S BIGGEST LIVING FISH' appears in the bottom left corner, with the number '1' in a yellow box, indicating its status as the world's largest fish. The text then changes to 'THE WORLD'S BIGGEST LIVING FISH IS A SHARK.' The scene transitions to a vibrant coral reef teeming with small fish, showcasing the rich biodiversity of the underwater world. The text 'The world's biggest living fish is a shark' appears in the bottom left corner, reinforcing the information. The clip concludes with a diver swimming near the whale shark, emphasizing the scale of the creature in comparison to the diver.

In the audio, a woman's voice states, "The world's biggest living fish is a shark. Of the estimated 34,000 species of fish, the largest are whale sharks." Soft background music accompanies her speech, enhancing the serene atmosphere.

The video then transitions to a diver swimming near the whale shark, which is now much closer, showcasing its massive size. The diver, equipped with a scuba suit and fins, moves slowly through the water, highlighting the scale of the whale shark. The scene shifts to a closer view of the whale shark, revealing its distinctive white spots and immense size as it swims gracefully through the clear blue ocean. The text 'Shark' appears in the bottom right corner, emphasizing the species of the whale shark. The clip concludes with a close-up of the whale shark's mouth, showcasing its gills and the intricate details of its anatomy.

In the audio, the woman continues, "These gentle giants usually grow to about 40 feet long and weigh an estimated 15 tons. Their mouths alone can span 4 feet wide." The background music continues to play, maintaining the tranquil ambiance.

The final clip features a close-up view of the whale shark swimming gracefully in the deep blue ocean. The whale shark's distinctive white spots are clearly visible as it moves slowly through the water. The text 'clideo.com' appears in the bottom right corner, indicating the source of the video. The scene transitions to a closer view of the whale shark's mouth, showcasing its gills and the intricate details of its anatomy. The whale shark's mouth is open, revealing its mouth and teeth, as it swims slowly through the water. The clip concludes with a close-up of the whale shark's mouth, emphasizing its size and the surrounding marine life.

In the audio, the woman's voice concludes, "Their mouths alone can span 4 feet wide." The background music continues to play, maintaining the serene atmosphere.

Overall, the video provides a comprehensive visual and auditory exploration of the whale shark, highlighting its size, distinctive features, and the rich biodiversity of the underwater world.

Figure 8: A caption generated by the model before mrDPO. Red parts are texts described incorrectly.

**Output by the model after 3 gDPO rounds:**

The video begins with an underwater scene where a large shark is swimming gracefully through the deep blue ocean. The shark's body is covered in a pattern of white spots, and it moves elegantly among schools of small fish. As this visual unfolds, text appears on the screen: 'THE WORLD'S BIGGEST LIVING FISH' in a yellow box, with the number '1' prominently displayed in a larger yellow box below it. This text is accompanied by a woman's voice-over: "The world's biggest living fish is a shark. Of the estimated 34,000 species of fish, the largest are whale sharks." The scene transitions smoothly to another part of the ocean where a vibrant coral reef comes into view, teeming with small, colorful fish swimming among the intricate structures of the reef. The focus then shifts to a diver in a scuba suit swimming alongside the massive shark, emphasizing the sheer size of the creature in comparison to the diver.

The narrative continues with the diver in a scuba suit swimming alongside the large shark, which remains the focal point as it moves gracefully through the deep blue ocean. The camera angle changes to show the shark from behind, revealing its impressive size and distinctive white spots pattern. The diver stays close, underscoring the scale of the shark in comparison to the human. The woman's voice continues: "These gentle giants usually grow to about 40 feet long and weigh an estimated 15 tons. Their mouths alone can span 4 feet wide..." The soft background music continues to enhance the serene underwater environment. The clip then transitions to a close-up view of the shark's head and mouth, with the shark swimming slowly, its mouth opening and closing, surrounded by the small fish that accompany it. The clip concludes with the shark's head prominently in view, emphasizing its grandeur and the intricate details of its features.

The video seamlessly blends visual and auditory elements to convey the awe-inspiring scale and beauty of the world's biggest living fish, the whale shark, in its natural habitat.

Figure 9: A caption generated by the model after 3 gDPO rounds. Red parts are texts described incorrectly. Some details lack in the video caption, such as Event 12 in Figure 7.

**Output by the model after 6 gDPO rounds:**

The video is a captivating underwater journey featuring a majestic whale shark, accompanied by detailed audio-visual descriptions that enhance the viewing experience.

From a text overlay on the screen, it is indicated that the video is about the world's biggest living fish, which is a shark. The overlay shows a yellow box with the text 'THE WORLD'S BIGGEST LIVING FISH IS A SHARK.' aligning with the narration. As the scene transitions to a diverse coral reef teeming with small fish, the camera showcases the rich marine life in the background. The visual narrative continues with a diver in a diving suit and an oxygen tank swimming alongside a massive whale shark, emphasizing the sheer size and grace of the creature in the deep blue ocean. The text 'clipdo.com' is visible in the top right corner of the frames, indicating the source of the video.

The audio aligns with the visuals by providing context: "The world's biggest living fish is a shark. Of the estimated 34,000 species of fish, the largest are whale sharks." This narration is accompanied by soft background music, enhancing the grandeur of the subject matter.

The underwater journey progresses with the diver in a diving suit and oxygen tank swimming alongside the massive whale shark, which is shown in greater detail, revealing its distinctive pattern of white spots and immense size. The clear blue water and visible coral reef in the background enhance the visual spectacle. The camera remains focused on the whale shark, capturing its slow and deliberate movements as it swims through the water. The clip transitions smoothly to a frontal view of the whale shark's head and mouth, with its gills and part of its body clearly visible, surrounded by small fish that swim in the same direction. The whale shark's mouth is open, showing its interior, and the small fish continue to swim around it, emphasizing the symbiotic relationship between the whale shark and the smaller fish. The clip concludes with this intimate view of the whale shark's head, highlighting its massive size and the delicate marine life that surrounds it. The text 'clipdo.com' remains visible in the top right corner, indicating the source of the video.

The accompanying audio continues to provide fascinating information: "These gentle giants usually grow to about 40 feet long and weigh an estimated 15 tons. Their mouths alone can span 4 feet wide." This description is accompanied by the sound of a whale's blow, adding to the awe-inspiring nature of the scene.

The video culminates with a frontal view of the whale shark's head and upper body, its distinctive white spots clearly visible against the deep blue background. The whale shark's mouth is open, showing its gills and part of its mouth interior, with numerous small fish swimming around it, emphasizing the symbiotic relationship. The whale shark's head remains the focal point as it continues to swim forward, with the small fish maintaining their presence around it. The clip concludes with a close-up of the whale shark's mouth, highlighting the intricate patterns and the numerous small fish that accompany it, showcasing the delicate and fascinating nature of the marine ecosystem. The text 'clipdo.com' remains visible in the top right corner, indicating the source of the video.

The audio aligns with the visuals by describing the whale shark's head and the small fish surrounding it, with a calm and informative tone: "The whale shark's head and mouth are clearly visible, emphasizing the intricate patterns and the small fish that accompany it." This description is accompanied by a soft underwater sound, enhancing the sense of tranquility and wonder of the underwater world.

The entire video cohesively presents the majestic and complex nature of the whale shark and the intricate marine ecosystem it inhabits, with detailed visual and audio descriptions that leave a lasting impression on the viewer.

This detailed description is intended to guide viewers through the various segments of the video, highlighting the key points and the fascinating world of the whale shark and its environment.

Figure 10: A caption generated by the model after 6 gDPO rounds. Red parts are texts described incorrectly.

**Output by the final video-SALMONN 2:**

The Video aligns with a detailed visual and auditory description of a shark, likely a whale shark, swimming gracefully through the deep blue ocean, accompanied by various underwater scenes and accompanying narration.

The video starts with a majestic whale shark swimming gracefully through the deep blue ocean, surrounded by small fish in the background, with text overlay appearing on the screen that aligns with the narration: 'THE WORLD'S BIGGEST LIVING FISH IS A SHARK.' This text emphasizes the sheer size and importance of the whale shark in the context of the world's largest living fish. The scene transitions to show a diverse coral reef teeming with small fish, indicating the rich biodiversity of the marine environment. The focus then returns to the whale shark, now accompanied by a diver in a scuba suit, indicating a sense of scale and human interaction with this magnificent creature.

The audio starts with a man's voice stating, "The world's biggest living fish is a shark. Of the estimated 34,000 species of fish, the largest are whale sharks." This statement sets the stage for the viewer, highlighting the specific focus of the video on whale sharks. The background music adds a sense of grandeur and emphasizes the scale and importance of these creatures.

The narrative continues with the whale shark and the diver in the deep blue ocean, with the diver swimming alongside the massive fish, showcasing the awe-inspiring size difference. The whale shark's distinctive pattern of white spots and patches on a greyish background is clearly visible, and the serene underwater environment with visible coral structures enhances the visual experience. The diver remains in the frame, emphasizing the scale and the human interest in observing such a magnificent creature up close.

The audio description complements this by stating, "These gentle giants usually grow to about 40 feet long and weigh an estimated 15 tons." This information provides specific details about the whale shark's size and mass, reinforcing the awe and wonder of the creature. The man's voice is accompanied by the sound of water splashing, adding to the immersive experience.

The video then transitions to a close-up view of the whale shark's head and upper body, with its mouth open, revealing the baleen plates inside, surrounded by small fish that enhance the sense of scale and community. The whale shark swims forward, and the fish swim around its mouth, showcasing the intricate ecosystem and the whale shark's role within it. The deep blue background remains consistent, reinforcing the underwater setting and the tranquility of the environment.

The accompanying narration continues with the man explaining, "Their mouths alone can span four feet wide," providing additional context about the whale shark's mouth size and emphasizing the sheer scale of these magnificent creatures. The music remains consistent, enhancing the grandeur and the importance of the whale shark's presence in the marine ecosystem.

The entire video cohesively presents a stunning visual and auditory journey into the world of the whale shark, highlighting its size, beauty, and the intricate underwater ecosystem it inhabits. The use of text overlays and narration guides the viewer through the information, ensuring a comprehensive understanding of the subject matter.

This description is intended to provide a detailed and informative summary of the video, guiding the viewer through the majestic journey of the whale shark and the rich biodiversity of the marine environment.

Figure 11: A caption generated by the final video-SALMONN 2 model, which is obtained after one round of gDPO training following rebirth tuning. Red parts are texts described incorrectly.

**An example of the unnatural video caption:**

Video Transition Description:

[Video Transition Description continues:]

The video starts with a close-up view of a shiny silver sports car, likely a high-end model given its sleek design and the context provided later in the video. The car's hood and part of the windshield are visible, reflecting the surrounding environment which includes other vehicles and a backdrop of tall evergreen trees, indicating a likely outdoor or semi-outdoor setting, such as a parking lot or a car show. A person wearing a red cap and dark clothing is present, and it seems they are speaking directly to the camera, likely providing details or a review about the car. As this happens, the audio reveals the person saying:

- "Made it to a six-speed manual transmission. I'll have more on the horsepower and torque in just a little bit. The engine's running..." This is followed by the sound of the engine revving multiple times, adding a dynamic feel to the presentation and setting the stage for the car's specifications.

The scene then transitions smoothly into a detailed view of the same silver sports car from various angles, emphasizing its sleek design and features such as the fuel cap labeled 'VESTER' and the rear lights. The car's license plate is visible but should not be read aloud to respect privacy.

The focus remains on the car's aesthetic and engineering details, then the scene shifts to the car's interior, revealing two individuals seated in the car. Both are wearing seat belts and appear to be engaged in a conversation, likely delving into the car's performance or the driving experience. The background through the car's windows shows a blurred view of the outside, indicating motion or a change in scenery as the car is in motion, likely on a test drive or a drive-through a scenic route.

Concurrently Audio Description:

The conversation continues with the man detailing the car's specifications: "So I so it's a 8.4 liter B10 six-speed manual what's the horse and torque on it 645 horsepower 600 and 600 pound-pounds of torque wow and at a very low rp..." This description provides specific technical details about the car's engine performance, matching the visual emphasis on the car's power and performance.

The clip then develops through the car's interior setting, showing the two individuals still conversing. The text '@IMDAVICKSON' and '@EVERYMANDRIVER' appears at the top of the frames, indicating social media handles likely of the individuals or entities presenting the car. This is followed by text at the bottom of the frames reading 'Scott Brown, Dodge' and 'EVERYMANDRIVER.COM - News, Reviews, Videos,' which offers context and credit to the individuals or entities providing the review. The background through the car windows continues to show a blurred view of the outdoors, confirming the car is in motion on a scenic route.

The conversation between the two individuals persists, with one holding a water bottle and the other gesturing animatedly, suggesting a dynamic and engaging discussion about the car's performance and the driving experience. This is complemented by the man's voice remarking:

- 'One of the things...' which hints at a broader discussion likely delving into the car's features and the enjoyment derived from driving it, enhancing the viewer's understanding of the car's capabilities and the pleasant experience it offers.

The video cohesively presents both exterior and interior perspectives of the car, along with detailed audio information about the car's specifications and the driving experience, providing a comprehensive review for the audience.

Video Description Concluding:

This final segment ensures the viewer is given a comprehensive and detailed look at the silver sports car, along with the engaging narrative about its performance and driving experience. The seamless transition between exterior and interior views, along with the continuous dialogue about the car's features, helps to deliver a thorough and engaging review.

-- This description meets the video's structure while providing a coherent and detailed narrative that connects all the elements presented, giving the audience a comprehensive understanding of the car and the experience it offers.

The video meets audience expectations by delivering both visual and auditory details, making it an informative and engaging watch for those interested in the car and its features.

Text Description Translation:

This description is in Chinese, and the translation is provided to ensure the same detailed and comprehensive information is conveyed to the audience in both languages.

This description meets the expectations set by the structure and content of the video, delivering a thorough review and detailed description of the car's features and performance.

--'--'

[Video Description END]

The video description is intended to be informative and engaging, ensuring viewers gain a comprehensive understanding of the silver sports car's features and performance through a detailed presentation of both exterior and interior views, along with insightful audio commentary.

--'--'

Figure 12: A very unnatural caption generated by the model after too many DPO rounds, which includes strange characters and sentences, like "[Video Transition Description continues:]" and "–".

