# OpenReview forum: "Enhancing Multimodal LLM for Detailed and Accurate Video Captioning using Multi-Round Preference Optimization"
_ICLR.cc/2025/Conference — Submitted to ICLR 2025_

### Official Review · Reviewer_aYgd · 2024-10-23

**Soundness:** 3
**Presentation:** 2
**Contribution:** 2
**Rating:** 6
**Confidence:** 5

**Summary:**

This paper presents a large multimodal model (audio-visual) for video captioning task. multi-round DPO and LoRA is used to train the model. Experiments look good, showing that the training recipe enhances video-SALMONN 2’s captioning accuracy. Besides, this paper introduce an evaluation pipeline that computes the missing and hallucination rates of audiovisual events in video captions using text-based LLMs, breaking down the process into sub-tasks suited for current LLMs and a new benchmark for video captioning with a human-annotated test set.

**Strengths:**

1. Audio is very important in understanding video, and incorporating it is a great idea.
2. The approach of using DPO (Direct Preference Optimization) and LoRA (Low-Rank Adaptation) to enhance specific capabilities with limited data is intuitive and effective.
3. Detailed video captioning is an excellent task, although it is not novel. It's great to see that people are starting to pay attention to and evaluate this task.
4. The event missing rate (Miss), hallucination rate (Hall), and text repetition rate (Rep) are introduced as the new metric for evaluation the detailed video captioning tasks.
5. Dividing the tasks into global and local parts is very beneficial. A recent work, MovieChat [1] also made a similar categorization (for video QA). Perhaps the authors could discuss and compare this in future versions.

[1] Song, Enxin, et al. "Moviechat: From dense token to sparse memory for long video understanding." Proceedings of the IEEE/CVF Conference on Computer Vision and Pattern Recognition. 2024.

**Weaknesses:**

1. The writing and illustrations in the paper need improvement, especially the figures, which severely hinder the proper understanding of the article.
2. Since DPO was used for training, based on my experience, DPO does not offer as stable training conditions as SFT. I hope the authors can present the specific parameters and training curves for DPO to further enhance reproducibility.
3. Although the authors present LoRA as one of the main contributions, there is no ablation study to demonstrate how LoRA prevents catastrophic forgetting or performance drop.
4. I believe that although the automated metrics are good (and perhaps better than metrics like CIDEr), the new metrics still need to align with human perception. I suggest that the authors refer to a concurrent work, AuroraCap [1], where they first use human Elo to score each model and then calculate the correlation between human Elo and the proposed metrics in the paper. This would make the benchmark more convincing.
5. There are too few details about the proposed benchmark. I only found statistics on video types and lengths in the supplementary material, but there is no clear information regarding caption length (word count, noun count) or the number of video sources.

[1] Chai, Wenhao, et al. "AuroraCap: Efficient, Performant Video Detailed Captioning and a New Benchmark." arXiv preprint arXiv:2410.03051 (2024).

**Questions:**

Please revise the Strengths and Weaknesses sections point by point. This is a paper with great potential. If the authors can provide additional responses to certain issues, discuss related work more thoroughly, and include more experiments and observations, I would be very happy to raise my score.

Additionally, although this is a concurrent work (and the authors can choose not to compare, as it's not a mandatory requirement), I would be very interested to see the model's performance on the VDC benchmark [1]. This would help me make a final decision. Again, this is completely optional.

[1] https://huggingface.co/datasets/wchai/Video-Detailed-Caption

---

> ### Author Response · Authors · 2024-11-16
> **Response to Reviewer aYgd**
>
> Thank you very much for your thoughtful and constructive feedback on our paper. We sincerely appreciate the time and effort you dedicated to reviewing our work, and will revise our paper carefully according to your valuable suggestions. Below, we will respond to each of your points in detail.
>
> - **Weakness 1**
>   - We apologise for the lack of clarity. We will revise and improve the clarity of this paper carefully according to the valuable feedback and suggestions from the reviewers.
>
> - **Weakness 2**
>
>   - In DPO training, the learning rate is 5e-6. We use 64 GPUs to train the model and the batch size per GPU is 1. The coefficient $\lambda$ of gDPO in Eqn. 11 in the paper is set to 0.1. Actually, a suitable learning rate is important for DPO training. We have searched for a lot of learning rates: 5e-5, 2e-5, 1e-5, 5e-6, 2e-6 and 1e-6, and 5e-6 works best. Too large or too small learning rate may result in training failure. We will include all these details in the revision of our paper. We also promise to release all our datasets, source code and model checkpoints to achieve full reproducibility of our paper, if it is accepted.
>   - The DPO loss is initially about 0.8 and almost reaches zero in the end, while cross entropy loss for the ground truth fluctuates around 1.5. We will provide the training curves in our revised paper.
>
> - **Weakness 3**
>
>   - There may be a confusion here. LoRA Proxy doesn't prevent catastrophic forgetting. It's rebirth tuning that alleviates the problem.
>
>     Actually, the LoRA proxy is a more beneficial method for performance improvement in mrDPO compared to directly training the same LoRA, especially after over 4 rounds of gDPO.
>
> - **Weakness 4**
>
>   - Thanks for your advice. We will conduct the Elo test for our benchmark and add the analysis results to the revised paper. We're now trying our best to conduct the Elo test and provide the results to the reviewers as soon as possible.
>   - Besides, we conduct a case study at https://github.com/Video-SALMONN-2/Video-SALMONN-2.github.io/blob/main/rebuttal/README.md to prove the performance of our method.
>
> - **Weakness 5**
>
>   - The details of our test set are given below:
>
>   - | Video Source | #Sample | Average Duration | #Vocab. | #Word   | Average Caption Length | Average Atomic Event Num. |
>     | ------------ | ------- | ---------------- | ------- | ------- | ---------------------- | ------------------------- |
>     | Youtube      | 483     | 51 seconds       | 17137   | 296,938 | 615 words              | 34                        |
>
>   -  We will include these details in our revised paper.
>
> - **Weakness 6**
>
>   - The captioning results of our Video SALMONN 2 model and the AuroraCap model (SOTA model on the VDC benchmark) are shown below. All the results are tested by gpt-4o-08-06.
>
>   - |                      | Camera (Acc%/Score) | Short (Acc%/Score) | Background (Acc%/Score) | Main Object (Acc%/Score) | Detailed (Acc%/Score) |
>     | -------------------- | ------------------- | ------------------ | ----------------------- | ------------------------ | --------------------- |
>     | AuroraCap (7B)       | **43.5/2.27**       | **32.07/1.68**     | 35.92/1.84              | 39.02/1.97               | **41.30/2.15**        |
>     | Video-SALMONN 2 (7B) | 35.81/1.94          | 28.77/1.52         | **40.83/2.13**          | **40.67/2.12**           | 36.67/1.96            |
>
>     | Task                 | VDC Score (Acc%/Score) |
>     | -------------------- | ---------------------- |
>     | AuroraCap (7B)       | **38.21/1.98**         |
>     | Video-SALMONN 2 (7B) | 36.55/1.93             |
>
>   - As shown in the tables, on the VDC benchmark, our Video-SALMONN 2 model can achieve competitive results compared with the AuroraCap model, a contemporary work on detailed video caption (without audio) with SOTA performance on the VDC benchmark.
>
>      Note that the VDC benchmark is a pure visual benchmark, which means the videos in the VDC benchmark are all videos without any audio. Since Video-SALMONN 2 was not trained on any silent video data in the SFT and mrDPO stages, the video captions it produces for VDC are potentially not of the best quality. We believe video-SALMONN 2 can generate better captions if there are paired audio tracks that come along with the videos in the VDC benchmark. We observed that video-SALMONN 2 generates severe hallucinations in the captioning silent video. Here are a few examples. Although this could be resolved by retraining the model with silent video, the goal of this paper is to study videos with paired audio.
>
>     ```JSON
>     {
>         "question": "How are the different viewpoints presented?",
>         "answer": "through frequent angle shifts",
>         "pred_answer": "Through visual actions and continuous audio commentary."
>     },
>     {
>         "question": "Who or what is guiding the individual through the crafting process",
>         "answer": "manual",
>         "pred_answer": "The man's voice in the audio."
>     },
>     ```

---

> > ### Comment · Reviewer_aYgd · 2024-11-16
> >
> > Great rebuttal! As a reviewer, I appreciate your efforts.
> >
> > For Q1, Q6, well done. Q3 is a interesting trick, I buy it.
> >
> > Since I'm not specialized in audio, so I'd like to know if, when applying audio to video tasks, there are other aspects beyond human speech/dialogue that can provide significant benefits (considering the additional training and inference cost). so for Q5, would you mind to provide some details about the audio part. For example, what is the proportion of human voices, and how much of it consists of off-screen audio (where the source of the audio is not visible in the video)? You don’t need to provide me with exact statistics; I’m just looking for a rough estimate.
> >
> > For Q2 and Q4, please update the context in the appendix and indicate the page numbers in your follow-up response. I assure you that I will promptly raise my score.

---

> ### Author Response · Authors · 2024-11-16
> **Response to Reviewer aYgd**
>
> Thank you very much for your quick response!
>
> In most cases, non-speech events play a role in assisting video understanding, where the information of the video is mainly derived from visual and speech. But in some cases, non-speech events are also very necessary, such as the laughter of the audience (outside the video), the sound of car horns (outside the video), and so on.
>
> By string matching, we crudely count speech-related and non-speech atomic events for all videos in the test set. The speech-related events are about the human voice, and the non-speech events are like music, wind blowing, bird chirping et al. The rough statistics are listed below. On average:
>
> - Each video contains **34.2** atomic events.
> - Within these, there are **6.1 audio-related** atomic events per video, including **4.6 speech-related** events and **1.5 non-speech** events.
>
> The visual element provides more information than the audio element, but the latter also accounts for a certain portion, especially the human voice.
>
> For off-screen audio, there are a lot of cases. For example, this is a video in our benchmark: https://www.youtube.com/watch?v=t3Y2eYBos4k
>
> Sometimes the speaker appears out of frame or does not show his full body within the video. And non-speech events like the music are off-screen.
>
> It is hard to detect off-screen audio events in the whole benchmark. We just manually and randomly watched 10 videos. We estimate that about 10%-20% of the speech content is said by speakers outside the video frames. For non-speech events, human-related events such as children crying or people screaming are usually in the video, while other events like music are always off-screen.
>
> For Q2 and Q4, we will update the paper later.

---

> > ### Comment · Reviewer_aYgd · 2024-11-16
> >
> > Thanks for your information. I think it would be great to include all those details in the updated draft. Good luck for the rest of your rebuttal.

---

> ### Author Response · Authors · 2024-11-23
> **Response to Reviewer aYgd**
>
> We have uploaded the revised paper. The revised parts are marked blue.
>
> **For Q2:**
> The training curves are provided in **Appendix G (Lines 878-907), and a pointer in Lines 384-385.**
>
> **For Q4:**
> We conduct Elo rating to rank the video captioning capability of Gemini-1.5-pro, GPT-4o, our visual base model, and the final video-SALMONN 2. Parameters of the Elo rating system are provided below:
>
> | **Parameter**     | **Value** |
> | ----------------- | --------- |
> | Initial ELO mean  | 1000      |
> | Base of logarithm | 10        |
> | Scaling factor    | 400       |
> | K-factor          | 2         |
>
> We collected 360 pairs of captions generated by different models as simulated matches, where each model generates 180 video captions. Annotators are provided with the video and two corresponding video captions generated by two models, respectively, and they are expected to select the better caption based on its completeness and correctness.
>
> Compared with the Elo experiments in AuroraCap, we use a smaller K-factor. This is because we use a much smaller number of simulated matches (360 vs 2778) due to the time constraint of the rebuttal period and each model wins multiple times against any other model. Therefore, we cannot increase the K-factor, otherwise the rating will not converge. We repeated the collected comparison 8 times to form 2880 matches to get stable rating results. The final Elo rating results and the global caption error rates of each model are provided below:
>
> | **Model**        | **Total** Error Rate% ↓ | **ELO** Rating ↑ |
> | ---------------- | ----------------------- | ---------------- |
> | GPT-4o Visual    | 33.8                    | 1048             |
> | Gemini-1.5-pro   | 38.3                    | 1010             |
> | Ours-Visual Base | 59.7                    | 888              |
> | video-SALMONN 2  | **13.7**                | **1054**         |
>
> The results are sufficient to demonstrate the consistency between our proposed automatic metric and human perception. We have updated the paper with the discussion and results in **Appendix H (Lines 910-940), and a pointer in Lines 430-431.** We are still conducting larger-scale Elo experiments and will update the results once completed.
>
> The results on the VDC benchmark are also provided in **Table 9 (Lines 983-995).**

---

> > ### Comment · Reviewer_aYgd · 2024-11-23
> >
> > Thanks for your final results. I’ve updated my score. I believe those exp results would make this paper much stronger. Have a good time.

---

> > > ### Author Response · Authors · 2024-11-23
> > > **Response to Reviewer aYgd**
> > >
> > > Thank you for updating your rating of our paper from 5 to 6—we truly appreciate your recognition of our revisions. To further improve the paper, we’d be grateful for any additional feedback or suggestions you might have that could help us address any remaining concerns and potentially raise the rating further.
> > >
> > > Thank you for your time and consideration.

---

### Official Review · Reviewer_7DpV · 2024-11-02

**Soundness:** 2
**Presentation:** 3
**Contribution:** 2
**Rating:** 5
**Confidence:** 4

**Summary:**

This paper introduces video-SALMONN 2, a powerful audio-visual large language model (LLM) designed for detailed and accurate video captioning. The model is built upon a pre-trained visual LLM and is further enhanced with auditory capabilities by training on audio-only data and videos with synchronized audio tracks. To accurately assess the model's performance, new metrics are proposed to evaluate the completeness and accuracy of video descriptions, which are then optimized using a novel multi-round direct preference optimization (mrDPO) approach. The mrDPO process involves periodically updating the DPO reference model, merging and re-initializing the low-rank adaptation (LoRA) module, and incorporating guidance from ground-truth video captions to stabilize the training. To address potential catastrophic forgetting of non-captioning abilities, a rebirth tuning process is introduced, which fine-tunes the pre-DPO LLM using the captions generated by the mrDPO-trained model as supervised labels.

**Strengths:**

1.the proposed novel pipeline including mrDPO, LoRa proxy and rebirth, is detailedly introduced, and experiment results on each iteration are well presented.
2.good ablation experiments are provided on each mrDPO iteration, LoRa proxy, and rebirth. And the experiments provide good observation on the DPO' effect on the model performance.
3.the whole pipeline is interesting and inspirational.

**Weaknesses:**

1.the model performance improvement is mainly evaluated on the internal benchmarks, and the quality of the internal benchmark is unknown.
2. video-mllms could not only do recgnition and captioning of a video, but also exploite the knowledge of LLM to do further recognition reseasoning. However, in paper, only one comparison with public benchmark "Video-MME short video" is provided. So it is not sure the proposed method could be beneficial for comprehensive improvements of a video-mllm.

**Questions:**

1. for video QA evaluation，could authors provide result on the Video-MME Medium and Long Video track? and also ablation results on other video understanding benchmark, such as mvbench, videovista, MLVU, to make the comparison more transparent and persuasive?
2. for video captioning evaluation, could authors provide more comparison results on  public benchmark, such as MSR-VTT and Vatex？
3. will author plan to publicly release the internal benchmark?

---

> ### Author Response · Authors · 2024-11-16
> **Response to Reviewer 7DpV**
>
> Thank you very much for your thoughtful and constructive feedback on our paper. We sincerely appreciate the time and effort you dedicated to reviewing our work, and will revise our paper carefully according to your valuable suggestions. Below, we will respond to each of your points in detail.
>
> - **Weakness 1**
>
>   - This internal benchmark is carefully checked by humans, and the evaluation pipeline is also manually checked. Therefore, we can ensure the high quality of the internal benchmark. We also promise to release all of our datasets, source code and model checkpoints, if the paper is accepted.
>
> - **Weakness 2 and Question 1**
>
>   - When processing with audio and video, we sample 1 frame per second for visual input and utilize the entire audio track for audio input. As a result, our model struggles to handle long videos with paired audios, since the use of the complete audio track leads to an excessively long audio token sequence that can exceed the memory limit. That is why we only evaluate the VideoMME Short subset. In comparison, most public video LLMs evaluated on these datasets do not have paired audio input.
>
>     To avoid this difficulty, to test VideoMME "Medium" and "Long" sets, we discard the audio track and only input the model with the video frames. Besides VideoMME, we also test NeXT-QA, MVBench and VideoVista. These test sets are all video QA benchmarks. For VideoVista, we also discard audio for overly long video when testing.
>
>     The QA results are shown as follows. Note that the training set of VideoChat 2 includes NeXT-QA training data, while video-SALMONN 2 did not use any of these datasets during training.
>
>     | Model                    | VideoMME Overall Acc% | NeXT-QA Acc% | MVBench Acc% | VideoVista Acc% |
>     | ------------------------ | --------------------- | ------------ | ------------ | --------------- |
>     | Video-LLaVA (7B)         | 39.9                  | -            | -            | 53.8            |
>     | Long-LLaVA (9B)          | 52.4                  | -            | 54.6         | -               |
>     | VideoChat 2 Mistral (7B) | 42.3                  | **78.6**     | **61.3**     | 54.9            |
>     | video-SALMONN 2 (7B)     | **54.1**              | 71.4         | 51.4         | **65.3**        |
>
>     We compare our model with other video LLMs that are of a similar size. As shown in the table, our model gets competitive results on several video QA benchmarks, indicating a relatively strong ability of video understanding. Specifically, we can get relatively high scores on VideoMME and VideoVista, which include paired audio-video data, which demonstrates the advantages of audio-visual LLM.
>
> - **Question 2**
>
>   - We test our model on the MSR-VTT data and count the length of our model's captions. Unfortunately, the average length of our model's captions is 527 words, while the average of the labelled captions is only 9 words, leading to both BLEU-4 and CIDER results being less than 1. We think that it is not suitable to use these metrics and datasets designed for brief captions to evaluate the ability to generate detailed captions by our model.
>
>     Besides MSR-VTT and Vatex, we also test our model on the recent VDC benchmark as Reviewer aYgd suggested, which is designed for detailed caption evaluation. The captioning results are shown as follows:
>
>     |                      | Camera (Acc%/Score) | Short (Acc%/Score) | Background (Acc%/Score) | Main Object (Acc%/Score) | Detailed (Acc%/Score) |
>     | -------------------- | ------------------- | ------------------ | ----------------------- | ------------------------ | --------------------- |
>     | AuroraCap (7B)       | **43.5/2.27**       | **32.07/1.68**     | 35.92/1.84              | 39.02/1.97               | **41.30/2.15**        |
>     | Video-SALMONN 2 (7B) | 35.81/1.94          | 28.77/1.52         | **40.83/2.13**          | **40.67/2.12**           | 36.67/1.96            |
>
>     | Task                 | Average VDC Score (Acc%/Score) |
>     | -------------------- | ------------------------------ |
>     | AuroraCap (7B)       | **38.21/1.98**                 |
>     | Video-SALMONN 2 (7B) | 36.55/1.93                     |
>
>     Our model is able to outperform the SOTA results in two of the five categories of VDC. In terms of the final score, our model can achieve competitive results as well compared with the SOTA on VDC. This indicates the captioning capabilities of our model.
>
> - **Question 3**
>
>   - Yes, we will release the internal benchmark upon acceptance, as well as the code, model checkpoints, and training data.

---

> > ### Author Response · Authors · 2024-11-23
> > **Response to Reviewer 7DpV**
> >
> > We have uploaded the revised paper. The revised parts are marked blue.
> >
> > The results for Question 1 and Question 2 are provided in Appendix I (Lines 942-996), and the pointer is at Line 431.
> >
> > We are looking forward to your reply.

---

> ### Author Response · Authors · 2024-11-25
> **Response to Reviewer 7DpV**
>
> Dear Reviewer 7DpV,
>
> Thank you for your invaluable feedback on our paper. We submitted our detailed rebuttal a week ago, addressing all the concerns you raised. We greatly appreciate the time and effort you invest in the review process, and we would be grateful if you could take a moment to review our rebuttal.
>
> We hope our clarifications and revisions demonstrate the value and significance of our work. Thank you for considering our rebuttal, and we look forward to your feedback.
>
> Best regards,
>
> The Authors

---

> > ### Author Response · Authors · 2024-11-26
> > **Response to Reviewer 7DpV**
> >
> > Dear Reviewer 7DpV,
> >
> > Thank you for your invaluable feedback on our paper. We submitted our detailed rebuttal a week ago, addressing all the concerns you raised. We greatly appreciate the time and effort you invest in the review process, and we would be grateful if you could take a moment to review our rebuttal.
> >
> > We hope our clarifications and revisions demonstrate the value and significance of our work. Thank you for considering our rebuttal, and we look forward to your feedback.
> >
> > Best regards,
> >
> > The Authors

---

> > > ### Author Response · Authors · 2024-11-27
> > > **Response to Reviewer 7DpV**
> > >
> > > Dear Reviewer 7DpV,
> > >
> > > Thank you for your invaluable feedback on our paper. We submitted our detailed rebuttal a week ago, addressing all the concerns you raised. We greatly appreciate the time and effort you invest in the review process, and we would be grateful if you could take a moment to review our rebuttal.
> > >
> > > We hope our clarifications and revisions demonstrate the value and significance of our work. Thank you for considering our rebuttal, and we look forward to your feedback.
> > >
> > > Best regards,
> > >
> > > The Authors

---

> > > > ### Author Response · Authors · 2024-11-29
> > > > **Response to Reviewer 7DpV**
> > > >
> > > > Dear Reviewer 7DpV,
> > > >
> > > > Thank you for your invaluable feedback on our paper. We submitted our detailed rebuttal two weeks ago, addressing all the concerns you raised. We greatly appreciate the time and effort you invest in the review process, and we would be grateful if you could take a moment to review our rebuttal.
> > > >
> > > > We hope our clarifications and revisions demonstrate the value and significance of our work. Thank you for considering our rebuttal, and we look forward to your feedback.
> > > >
> > > > Best regards,
> > > >
> > > > The Authors

---

> > > > > ### Author Response · Authors · 2024-12-01
> > > > > **Response to Reviewer 7DpV**
> > > > >
> > > > > Dear Reviewer 7DpV,
> > > > >
> > > > > Thank you for your invaluable feedback on our paper. We submitted our detailed rebuttal two weeks ago, addressing all the concerns you raised. We greatly appreciate the time and effort you invest in the review process, and we would be grateful if you could take a moment to review our rebuttal.
> > > > >
> > > > > We hope our clarifications and revisions demonstrate the value and significance of our work. Thank you for considering our rebuttal, and we look forward to your feedback.
> > > > >
> > > > > Best regards,
> > > > >
> > > > > The Authors

---

### Official Review · Reviewer_MeEe · 2024-11-02

**Soundness:** 2
**Presentation:** 1
**Contribution:** 2
**Rating:** 3
**Confidence:** 4

**Summary:**

In this paper, the authors propose a VideoLLM named video-SALMONN2, which is capable of understanding both the video and audio modality. To train the model, a training strategy with multi-round DPO (mrDPO) and rebirth tuning is applied. mrDPO iteratively performs DPO training for multiple steps, while rebirth tuning improves the original model with captions generated from the model updated with mrDPO. The performance is evaluated on the custom dataset, in terms of repetition and event missing rate.

**Strengths:**

1. video-SALMONN shows competitive performance on multiple metrics on the custom dataset for evaluation.
2. The goal this paper is aiming on is a promising direction.

**Weaknesses:**

1. Lots of details about the method seem to be missing or not clearly stated, making it hard to understand the method.
    1. Training losses are not detailed. It can be inferred that for the audio alignment stage, speech recognition and audio training losses are applied, for the SFT stage (L231-238), video captioning loss is applied, and for the DPO stages, a DPO loss is applied. Still, it would be explicitly mentioned which loss is applied for each stage, as it was very confusing, especially in the case of the audio alignment stage, where the information about training loss is simply mentioned in L376-377. Also, since there exist multiple variants of DPO, it would be better to include the exact formulation of DPO in the equation.
    2. How are the captions for DPO sampled? It is only stated as “First, distinct video captions are sampled from the model’s output distribution, given the
    input video” in L268.
    3. What are the definitions of ‘basic atomic events’ (L289), ‘information missing and hallucination rates’ (L292), ‘global captions’ (L269), and ‘local captions’ (L293)?
    4. How is a problematic pattern detected when building training data for rebirth training, which is mentioned in L351?
    5. It seems that the final model (video-SALMONN) is a model trained with rebirth training with data generated with mrDPO, and is further trained with a single round of gDPO (L407), which is not mentioned in the method section. It would be better to state the final model in the methods sections for better understanding.
2. The advantage of mrDPO is confusing.
    1. If training with mrDPO results in “the model gradually begins to produce repetitive and unnatural text patterns in its responses” (L337), why is it required?
    2. Such claim in 2-(a) is also contradictory with the statement in L349, which says the final mrDPO model “excels at generating complete and accurate video descriptions”.
    3. Also, are there experimental results that back up the claim that the mrDPO model “demonstrates significant improvements in captioning” (L335)?
    4. Results in Fig.4-(a) seem direct-DPO with single-round DPO shows the lowest total error rate, which makes the advantage of mrDPO more doubtful. Authors claim that the dashed line indicates a ‘high frequency of unnatural text generation’, but how are they determined as so?
3. Missing experiments and analysis
    1. Results on more conventional datasets are required. Current evaluations are only done on custom datasets and a Video-MME short set, where the method is compared only with a few baselines, extensive comparison on more conventional benchmarks with competitive baselines is required.
    2. The motivation for using GPT-based ‘missing’ and ‘hallucinatory’ events as metrics is not clear. Ablation results on metrics, comparing proposed metrics and conventional captioning metrics comparing generation results with ground-truth captions can make the design choice more convincing.
    3. Some qualitative examples showcasing outputs of an initial model, a model trained with mrDPO, and the final model would help understand the actual behavior of each model and the effect of each component.
4. Incorporating audio modality into VideoLLMs has already been proposed [1, 2, 3], and the proposed model seems to have no significant difference compared to them in terms of audio inclusion.

**References**

[1] Yang et al., Vid2Seq: Large-Scale Pretraining of a Visual Language Model for Dense Video Captioning, CVPR 2023

[2] Wang et al., InternVideo2: Scaling Foundation Models for Multimodal Video Understanding, ECCV 2024

[3] Zhang, Li, and Bing, Video-LLaMA: An Instruction-tuned Audio-Visual Language Model for Video Understanding, EMNLP 2023 (demo track)

**Questions:**

Please refer to the weaknesses part.

Overall, as mentioned in the weaknesses part I think the paper needs large revision at this point. First, the methods part can be written in a better way to make readers understand the technical details and contributions of this paper more easily. Also, more experiments and analysis would make the proposed methods more convincing.

---

> ### Author Response · Authors · 2024-11-16
> **Response to Reviewer MeEe (Part I)**
>
> Thank you very much for your thoughtful and constructive feedback on our paper. We sincerely appreciate the time and effort you dedicated to reviewing our work, and will revise our paper carefully according to your valuable suggestions. Below, we will respond to each of your points in detail.
>
> **Weakness 1: Lots of details about the method seem to be missing or not clearly stated, making it hard to understand the method**
>
> - **Weakness 1.1**
>
>   - In the audio alignment and SFT stages, the "Cross Entropy Loss" is computed on the labelled texts based on "Next Token Prediction". In the audio alignment stage, the model is trained on speech recognition and audio captioning tasks.
>   - In the mrDPO stage, we apply a revised DPO loss (gDPO). The loss is computed on $L_{gDPO}(\pi_{\theta}; \pi_{ref}) = -E_{(x,y_{win},y_{lose})\sim {D}}\left[\log \sigma \left(\beta \log \frac{\pi_{\theta}(y_{win}\mid {x})}{\pi_{ref}(y_{win}\mid {x})} - \beta \log \frac{\pi_{\theta}(y_{lose}\mid {x})}{\pi_{ref}(y_{lose}\mid {x})}\right)\right]  + \lambda E_{(x, y_{gt})\sim D_{human}}\log \pi_\theta(y_{{gt}}|x)$
>
>   - Here, the first term is the classical DPO loss, where $\pi_{\theta}$ is the policy model, $\pi_{ref}$ is the reference model. $y_{win}$ and $y_{lose}$ are the preferred sample and the dispreferred sample for the input video $x$. The second term is the classical cross-entropy loss on the labelled captions. We will include this standard DPO loss in the revision of the paper.
>
> - **Weakness 1.2**
>
>   - During LLM generation, we apply top-p sampling to obtain video captions for one video. Top-p sampling, also known as nucleus sampling, is a technique used in text generation that selects from a subset of the most probable tokens based on a cumulative probability threshold p.
>
>     In our implementation, we set the top-p value to 0.9 and the temperature value of 1.0 when sampling captions, and we sample two captions for one video to construct the training set of DPO.
>
> - **Weakness 1.3**
>
>   - We used the following pipeline to get the atomic events of the video and calculate the metrics (shown in Appendix E):
>
>     First, we have the groundtruth caption the video. We use GPT-3.5 or GPT-4o to split the groundtruth caption of this video into several atomic events, where we use GPT-4o for the test set and GPT-3.5 for the RL training set.  Atomic events need to be simple and clear and describe only one particular event. Here are some examples of the atomic events:
>
>      ["Man in yellow shirt holding a clear plastic water bottle",  "Man pouring water into a tall, clear glass", "Backdrop with dark blue wall and starry pattern", "Audio mentions a steep bridge in Japan", ...]
>
>     Then, the list of atomic events and the caption to be evaluated are simultaneously fed into GPT-3.5, to find out if there exist any atomic events in the list that are missed or hallucinated (L850). Finally, the quotient between the number of missing or hallucination events and the number of all events in the video is the final missing or hallucination rate (L853).
>
>   - The global caption is to describe the entire video, and the local caption is to describe the video at a specific time interval (L275-278).
>
> - **Weakness 1.4**
>
>   - We manually observed the caption of the model's output on the test set and directly counted the problematic pattern of the model's output, and subsequently directly removed data containing the same pattern during rebirth tuning. The patterns are something like "[Video Description]: ...", "###Audio Description\n...",  and some non-English descriptions and strange characters that are unnatural. Here is a case of unnatural caption: https://github.com/Video-SALMONN-2/Video-SALMONN-2.github.io/blob/main/rebuttal/README.md#unnatural-case
>
> - **Weakness 1.5**
>
>   - The final model is indeed obtained by training another DPO after rebirth tuning. We'll make it clearer in the revised version.

---

> > ### Author Response · Authors · 2024-11-16
> > **Response to Reviewer MeEe (Part II)**
> >
> > **Weakness 2: The advantage of mrDPO is confusing.**
> >
> > - **Weakness 2.1**
> >
> >   - As shown in Figure 4 (a), using single-round DPO can reduce the error rate from 53.6% to 40.8%, while using multi-round DPO can reduce the error rate to 23.3% (without LoRA Proxy), 21.3% (with LoRA Proxy), and 16.3% (with LoRA Proxy and gDPO regularization). Here we summarise the insight of using multi-round DPO below.
> >
> >     If we only train single-round DPO, the model converges quickly and the final performance will not be good enough. This is due to the fact that the reference model is biased against the most recent policy model update in the DPO algorithm, leading to a large difference in the output distributions of the reference model and the policy model after long-time training (e.g. 1000 updates), which therefore no longer satisfies the premise that they are approximately the same in DPO.
> >
> >     However, in multi-round DPO (mrDPO), new data will be sampled from the model after the last DPO round, and the model after the last DPO round will be the reference model, which satisfies the DPO premise. Therefore, the performance of the model can be further improved in a new round of DPO.
> >
> > - **Weakness 2.2**
> >
> >   - We believe there is a confusion here. In terms of the events appearing in the video, the model is able to generate video captions that cover most of these events with relatively few hallucinations. That is what we mean "complete and accurate".
> >
> >     The model may generate some strange patterns after mrDPO, which refer to the case that some texts in the output caption are not fluent and unnatural. However, even when this happens, most of the atomic events in the caption are accurate and complete. Further, with our proposed rebirth finetuning approach, the model can generate detailed and accurate captions without any unnatural pattern.
> >
> > - **Weakness 2.3**
> >
> >   - Figure 4 (a) shows the **total error rates of the model** with respect to **the number of DPO rounds**. Take the "Direct-gDPO" line as examples. The total error rate only drops by about 10% after a single DPO round. However, after six DPO rounds, the total error rate drops by over 30%, showing a significant improvement.
> >
> > - **Weakness 2.4**
> >
> >   - First, as described in the caption, all curves in Fig.-4(a) are multi-round DPO results, and there are no single-round DPO results shown in this figure. Multi-round DPO is necessary to achieve error rates below 30%.
> >
> >     As for the first round of DPO, direct-DPO (here diret DPO means DPO without using LoRA proxy) works the best compared with other methods in Fig.-4(a). However, Direct-DPO will lead to a slower model performance improvement after multiple rounds of DPO training and faster occurrence of unnatural patterns. Therefore, Proxy-DPO is the preferred setting.
> >
> >     Since these unnatural text outputs have similar patterns, it is not difficult to detect them using rule-based algorithms. We have examined the output of the model and found that such unnatural patterns are very limited. Therefore, for a video caption, we can detect whether there are a large number of occurrences of patterns like "[Video Description]: ...", "###Audio Description..." to determine whether this caption is unnatural.
> >
> > **Weakness 3: Missing experiments and analysis**
> >
> > - **Weakness 3.1**
> >
> >   - When dealing with synchronised audio and video, we sample 1  frame per second as visual input and use the whole audio track as audio input. Therefore, when handling long video with paired audio track, the model fails to process very long audio streams due to GPU memory limit. This is the reason why we only tested the VideoMME Short subset. In comparison, most published video LLMs cannot process the audio track, and a fixed number of visual frames are often sampled regardless of the length of the video.
> >
> >     To test VideoMME Medium and Long sets, we only input the model with the video frames for evaluation and discard the audio information. Besides VideoMME, we also test NeXT-QA, MVBench and VideoVista, as Reviewer 7DpV suggested. These test sets are all video QA benchmarks. For VideoVista, we discard audio for too long video when testing.
> >
> >     The QA results are shown as follows.
> >
> >   - | Model                    | VideoMME Overall Acc% | NeXT-QAAcc% | MVBench Acc% | VideoVista Acc% |
> >     | ------------------------ | --------------------- | ----------- | ------------ | --------------- |
> >     | Video-LLaVA (7B)         | 39.9                  | -           | -            | 53.8            |
> >     | Long-LLaVA (9B)          | 52.4                  | -           | 54.6         | -               |
> >     | VideoChat 2 Mistral (7B) | 42.3                  | **78.6**    | **61.3**     | 54.9            |
> >     | video-SALMONN 2 (7B)     | **54.1**              | 71.4        | 51.4         | **65.3**        |
> >
> >     Note that the VideoChat 2 model is trained on NeXT-QA, so NeXT-QA is an in-domain task for it.

---

> > > ### Author Response · Authors · 2024-11-16
> > > **Response to Reviewer MeEe (Part III)**
> > >
> > > **Weakness 3: Missing experiments and analysis**
> > >
> > > - **Weakness 3.1 (Continued)**
> > >
> > >   - Although our model is optimised to generate high quality captions, it is still able to generate competitive results on these video QA benchmarks compared to SOTA open-source video LLMs with a similar size. Specifically, we can get relatively high scores on VideoMME and VideoVista, which include paired audio-video data.
> > >
> > >   - We also test our model on the recent captioning benchmark VDC as Reviewer aYgd suggested. The results are as follows:
> > >
> > >   - |                      | Camera (Acc%/Score) | Short (Acc%/Score) | Background (Acc%/Score) | Main Object (Acc%/Score) | Detailed (Acc%/Score) |
> > >     | -------------------- | ------------------- | ------------------ | ----------------------- | ------------------------ | --------------------- |
> > >     | AuroraCap (7B)       | **43.5/2.27**       | **32.07/1.68**     | 35.92/1.84              | 39.02/1.97               | **41.30/2.15**        |
> > >     | Video-SALMONN 2 (7B) | 35.81/1.94          | 28.77/1.52         | **40.83/2.13**          | **40.67/2.12**           | 36.67/1.96            |
> > >
> > >     | Task                 | Average VDC Score (Acc%/Score) |
> > >     | -------------------- | ------------------------------ |
> > >     | AuroraCap (7B)       | **38.21/1.98**                 |
> > >     | Video-SALMONN 2 (7B) | 36.55/1.93                     |
> > >
> > >   - Our model is able to outperform the SOTA results in two of the five categories of VDC. In terms of the final score, our model can achieve competitive results as well compared with the SOTA on VDC.
> > >
> > > - **Weakness 3.2**
> > >
> > >   - The motivation is to evaluate a detailed video caption, since conventional captioning metrics (e.g. BLEU and CIDER) fail to reflect the performance of a detailed video caption. For example, when evaluating our model on the MSR-VTT test set, the average length of the generated captions from our model is 527 words, while that of the ground-truth captions is only 9 words. Based on this disparity, the results of BLEU-4 and CIDER are both less than 1. Therefore, it is unreasonable to utilize metrics for short caption evaluation to evaluate detailed captions.
> > >
> > > - **Weakness 3.3**
> > >
> > >   - We conduct a case study for the SFT model, the model after 3 DPO rounds, the model after 6 DPO rounds and the rebirth model. The details are shown here: https://github.com/Video-SALMONN-2/Video-SALMONN-2.github.io/blob/main/rebuttal/README.md
> > >
> > >     This is a video sample that we randomly select without cherry picking.
> > >
> > > **Weakness 4**
> > >
> > > - **Incorporating audio modality into VideoLLMs has already been proposed [1, 2, 3], and the proposed model seems to have no significant difference compared to them in terms of audio inclusion.**
> > >
> > >   - Thank you for pointing out these works, and we will add them to the references in the revised paper. Besides, although the inclusion of audio is not the core novelty of this paper, we would like to illustrate how our approach differs from other work.
> > >
> > >   - (1) Vid2Seq [1] utilized the transcription of the speech instead of the original audio, which results in the model ignoring other information such as emotion, speaker characteristics, and pronunciations in the speech signal and also not being able to perceive non-speech audio events. Besides, the text transcription of speech content often does not include any time information, which results in asynchronous audio and visual information streams.
> > >
> > >     (2) InternVideo2 has the same issue with Vid2Seq.
> > >
> > >     (3) Video-LLaMA has an audio input stream. However, the audio and the visual tokens are directly connected to form the input tokens of LLM, which causes the audio and visual information not being temporally aligned. While for our model, we interleave the audio and the visual tokens to synchronize the audio-visual information. Furthermore, Video-LLaMA also does not understand speech input.

---

> > > > ### Author Response · Authors · 2024-11-23
> > > > **Response to Reviewer MeEe**
> > > >
> > > > We have uploaded the revised paper. The revised parts are marked blue.
> > > >
> > > > We are looking forward to your feedback.

---

> ### Author Response · Authors · 2024-11-25
> **Response to Reviewer MeEe**
>
> Dear Reviewer MeEe,
>
> Thank you for your invaluable feedback on our paper. We submitted our detailed rebuttal a week ago, addressing all the concerns you raised. We greatly appreciate the time and effort you invest in the review process, and we would be grateful if you could take a moment to review our rebuttal.
>
> We hope our clarifications and revisions demonstrate the value and significance of our work. Thank you for considering our rebuttal, and we look forward to your feedback.
>
> Best regards,
>
> The Authors

---

> > ### Author Response · Authors · 2024-11-26
> > **Response to Reviewer MeEe**
> >
> > Dear Reviewer MeEe,
> >
> > Thank you for your invaluable feedback on our paper. We submitted our detailed rebuttal a week ago, addressing all the concerns you raised. We greatly appreciate the time and effort you invest in the review process, and we would be grateful if you could take a moment to review our rebuttal.
> >
> > We hope our clarifications and revisions demonstrate the value and significance of our work. Thank you for considering our rebuttal, and we look forward to your feedback.
> >
> > Best regards,
> >
> > The Authors

---

> > > ### Author Response · Authors · 2024-11-27
> > > **Response to Reviewer MeEe**
> > >
> > > Dear Reviewer MeEe,
> > >
> > > Thank you for your invaluable feedback on our paper. We submitted our detailed rebuttal a week ago, addressing all the concerns you raised. We greatly appreciate the time and effort you invest in the review process, and we would be grateful if you could take a moment to review our rebuttal.
> > >
> > > We hope our clarifications and revisions demonstrate the value and significance of our work. Thank you for considering our rebuttal, and we look forward to your feedback.
> > >
> > > Best regards,
> > >
> > > The Authors

---

> > > > ### Author Response · Authors · 2024-11-29
> > > > **Response to Reviewer MeEe**
> > > >
> > > > Dear Reviewer MeEe,
> > > >
> > > > Thank you for your invaluable feedback on our paper. We submitted our detailed rebuttal two weeks ago, addressing all the concerns you raised. We greatly appreciate the time and effort you invest in the review process, and we would be grateful if you could take a moment to review our rebuttal.
> > > >
> > > > We hope our clarifications and revisions demonstrate the value and significance of our work. Thank you for considering our rebuttal, and we look forward to your feedback.
> > > >
> > > > Best regards,
> > > >
> > > > The Authors

---

> > > > > ### Author Response · Authors · 2024-12-01
> > > > > **Response to Reviewer MeEe**
> > > > >
> > > > > Dear Reviewer MeEe,
> > > > >
> > > > > Thank you for your invaluable feedback on our paper. We submitted our detailed rebuttal two weeks ago, addressing all the concerns you raised. We greatly appreciate the time and effort you invest in the review process, and we would be grateful if you could take a moment to review our rebuttal.
> > > > >
> > > > > We hope our clarifications and revisions demonstrate the value and significance of our work. Thank you for considering our rebuttal, and we look forward to your feedback.
> > > > >
> > > > > Best regards,
> > > > >
> > > > > The Authors

---

### Official Review · Reviewer_zUKj · 2024-11-04

**Soundness:** 3
**Presentation:** 3
**Contribution:** 3
**Rating:** 3
**Confidence:** 5

**Summary:**

- The paper introduces video-SALMONN 2, an advanced audio-visual large language model (LLM) for generating detailed and accurate video captions, surpassing models like GPT-4o and Gemini-1.5-Pro on this task.

- A new evaluation pipeline is presented, with metrics specifically designed to quantify missing information, hallucination, and repetition rates in audio-visual captions, enhancing the precision of video content descriptions.

- Multi-round Directed Preference Optimization (mrDPO) is developed to improve caption accuracy through reinforcement learning, periodically updating a reference model and using low-rank adaptation (LoRA) for smoother and more effective model training.

- Rebirth tuning is introduced to prevent performance degradation in non-captioning tasks post-mrDPO, using supervised fine-tuning on self-labeled data to maintain both captioning quality and broader LLM capabilities.

**Strengths:**

- The introduction of video-SALMONN 2 with its emphasis on synchronized audio-visual inputs for captioning represents a significant advancement. By integrating both visual and audio data, the model moves beyond purely visual or audio-focused LLMs, capturing richer and more contextually complete video descriptions. The novel use of multi-round Directed Preference Optimization (mrDPO) as a reinforcement learning technique to improve captioning quality is a noteworthy approach.

- The paper demonstrates high-quality experimental validation, showcasing the effectiveness of mrDPO and rebirth tuning. Through detailed metrics like missing information, hallucination, and repetition rates, the authors provide quantitative evidence of video-SALMONN 2’s superiority over leading commercial models like GPT-4o and Gemini-1.5-Pro in video captioning tasks. The thorough benchmarking against these models on multiple metrics adds robustness to the study’s findings.

- The paper is clear and methodically structured, making complex techniques like multi-round preference optimization accessible. Each step in the methodology is well-illustrated, from model architecture through training strategies, with detailed explanations of mrDPO and rebirth tuning. The clarity of the evaluation pipeline and the metrics for assessing captioning quality further aids in understanding the model's improvements.

**Weaknesses:**

- The paper introduces rebirth tuning to help the model retain non-captioning abilities, but the evaluation of this feature feels limited. Expanding the assessment to include the model’s performance on non-captioning tasks, like video question-answering, at each training stage could give a more comprehensive picture of its capabilities. This approach would better demonstrate how effectively rebirth tuning preserves broader language skills.

- The evaluation pipeline depends heavily on other large language models, such as GPT-3.5 and GPT-4, to assess metrics like missing information and hallucination rates. This reliance raises concerns about objectivity and reproducibility, as these models come with their own biases. Developing an alternative, self-contained evaluation method or comparing LLM-based assessments with human judgments could strengthen the evaluation’s reliability and transparency.

- The model’s performance evaluation relies mostly on internal datasets, which limits the generalizability and reproducibility of the results. Including open datasets or releasing the test set would allow the research community to more easily validate the findings and enhance the overall credibility of the work.

- The mrDPO technique is key to the model’s effectiveness, but the paper lacks a detailed analysis of its individual components, like LoRA updates, regularization terms, and the multi-round structure. Investigating how each component contributes to the model’s overall performance would provide insights into which aspects are most beneficial, potentially guiding future improvements and applications of mrDPO.

- The training strategy, particularly mrDPO and the use of several LoRA proxies, adds computational overhead without a clear analysis of the associated trade-offs. Quantifying the specific GPU memory, runtime, and computational load required for each training round would clarify the model’s resource demands. Additionally, identifying the point at which diminishing returns begin in mrDPO could shed light on the efficiency of each round, highlighting which may be resource-intensive with limited added benefit. Exploring alternative configurations for resource-limited environments and comparing high-resource to optimized setups would offer insights into how performance scales in different compute settings. This analysis would clarify the balance between computational cost and performance improvements, making the model more adaptable and accessible for varied research needs.

- Although the model is designed to process both audio and visual inputs, the paper provides minimal qualitative insight into how well it aligns these features in its captions. Including examples of successful synchronizations as well as cases where alignment fails would offer a clearer view of the model’s multi-modal understanding abilities.

**Questions:**

- Could the authors include more diverse evaluation tasks to assess the impact of rebirth tuning on non-captioning skills? For instance, evaluating on tasks like video question-answering at each stage of training could provide more insights into how rebirth tuning preserves or enhances the model’s broader language abilities.

- Considering the reliance on GPT-3.5 and GPT-4 for assessing metrics like hallucination rates, how have the authors addressed potential biases from these models in their evaluation pipeline?

- Could the authors clarify if there are plans to evaluate video-SALMONN 2 on open, widely-used datasets for multimodal captioning? Including results on open datasets or releasing parts of the internal test set would enhance the model's credibility. It would also allow for broader validation by the research community, helping others replicate and build upon the findings.

- Can the authors provide a detailed analysis of the impact of individual components of mrDPO, like LoRA updates, regularization terms, and the multi-round structure, on the overall performance? How does each element contribute uniquely to the training effectiveness?

- Given that mrDPO involves multiple LoRA proxies and additional training rounds, can the authors quantify the GPU memory, runtime, and computational load associated with each round? Is there an observed point where additional rounds yield diminishing returns? Common metrics are FLOPS, MACs, Latency and so on.

- While the model is designed for audio-visual input, the paper lacks qualitative insight into its success rate in synchronizing audio and visual elements. Could the authors provide examples or failure cases?

---

> ### Author Response · Authors · 2024-11-16
> **Response to Reviewer zUKj (Part I)**
>
> Thank you very much for your thoughtful and constructive feedback on our paper. We sincerely appreciate the time and effort you dedicated to reviewing our work, and will revise our paper carefully according to your valuable suggestions. Below, we will respond to each of your points in detail.
>
> - **Weakness 1 and Question 1**
>
>   - We test the models on two video question-answering benchmarks: NExT-QA and VideoMME. Since our model is trained on paired audio videos of around 1 minute, we test the models on the "Short" set of VideoMME to reflect changes in the model's QA capabilities more accurately.
>
>   - |                                   | NExT-QA | VideoMME Short |
>     | --------------------------------- | ------- | -------------- |
>     | Ours-Visual Base                  | 72.8    | 67.2           |
>     | Ours-DPO Round 3                  | 72.0    | 67.3           |
>     | Ours-DPO Round 6                  | 70.2    | 65.3           |
>     | Ours-Rebirth Tuning               | 71.1    | 67.6           |
>     | Ours-Rebirth Tuning + 1 Round DPO | 71.4    | 67.0           |
>
>   - In the first few rounds of mrDPO, the model's QA capability remained largely unchanged. However, after 6 rounds of DPO, performance on the QA benchmark dropped, although the model's caption became more complete. After rebirth tuning, the model regained performance on the QA benchmark.
>
> - **Weakness 2 and Question 2**
>
>   -  First, in order to ensure the stability and reproducibility of our test results, we have fixed the random seed and set the top_p and temperature to 0.1 and 0.0 respectively when requesting GPT models.
>
>   -  As for our evaluation pipeline, while this automated evaluation process may have inaccurate evaluations, we have manually checked each part of our evaluation pipeline and ensured the process is as reliable as possible through prompt engineering.
>
> - **Weakness 3 and Question 3**
>
>
>   - As promised in the paper abstract, we will open-source the test data, training data, model checkpoints and source code once the paper is accepted. As for the results on other open datasets, we test our model on the recent VDC benchmark [1] as Reviewer aYgd suggested. The captioning results are as follows:
>
>   - |                      | Camera (Acc/Score) | Short (Acc/Score) | Background (Acc/Score) | Main Object (Acc/Score) | Detailed (Acc/Score) |
>     | -------------------- | ------------------ | ----------------- | ---------------------- | ----------------------- | -------------------- |
>     | AuroraCap (7B)       | **43.5/2.27**          | **32.07/1.68**        | 35.92/1.84             | 39.02/1.97              | **41.30/2.15**           |
>     | Video-SALMONN 2 (7B) | 35.81/1.94         | 28.77/1.52        | **40.83/2.13**         | **40.67/2.12**          | 36.67/1.96           |
>
>   - |                      | Average VDC Score (Acc/Score) |
>     | -------------------- | ----------------------------- |
>     | AuroraCap (7B)       | **38.21/1.98**                |
>     | Video-SALMONN 2 (7B) | 36.55/1.93                    |
>
>   - The AuroraCap 7B model is the SOTA model on the VDC benchmark, and our model can achieve competitive results. Note that the VDC benchmark is a pure visual benchmark. Our model's performance on this benchmark is degraded by the lack of paired audio input since it is trained only on paired audio-visual data.
>
>      In addition to the captioning results, we also test our model on open video QA benchmarks including VideoMME, NeXT-QA, MVBench and VideoVista, as Reviewer 7DpV suggested:
>
>      When testing the "medium" and "long" subsets of VideoMME, since the audio signal is overly long and exceeds the limit, we discarded the audio signal and only provided uniformly selected video frames to our model. Similarly, when testing VideoVista, we discarded the audio when testing videos longer than 2 minutes. The QA results are:
>
>     | Model                    | VideoMME Overall Acc% | NeXT-QA Acc% | MVBench Acc% | VideoVista Acc% |
>     | ------------------------ | --------------------- | ------------ | ------------ | --------------- |
>     | Video-LLaVA (7B)         | 39.9                  | -            | -            | 53.8            |
>     | Long-LLaVA (9B)          | 52.4                  | -            | 54.6         | -               |
>     | VideoChat 2 Mistral (7B) | 42.3                  | **78.6**     | **61.3**     | 54.9            |
>     | video-SALMONN 2 (7B)     | **54.1**              | 71.4         | 51.4         | **65.3**        |

---

> ### Author Response · Authors · 2024-11-16
> **Response to Reviewer zUKj (Part II)**
>
> - **Response to Weakness 3 and Question 3 (Continued)**
>
>   - Although the training data of VideoChat 2 includes the NeXT-QA training set, video-SALMONN 2 was not trained using any of these datasets, video-SALMONN 2  is still able to achieve competitive results on these video benchmarks.
>
>     Regarding using the MSR-VTT benchmark, we believe the BLEU and CIDER metrics often used for MSR-VTT are unsuitable for evaluating video-SALMONN 2. This is because the detailed captions generated by video-SALMONN 2 have an average length of 527 words, while the reference captions provided by the MSR-VTT benchmark have an average length of only 9 words.
>
> - **Weakness 4 and Question 4**
>
>   - The detailed analysis is shown in Sec. 5.2, and the results are shown in Figure 4 (a). Here we summarise the results in the table below.
>
>   - | Model                                                 | Total Error Rate% |
>     | ----------------------------------------------------- | ----------------- |
>     | Model before DPO                                      | 53.6              |
>     | Single-round DPO                                      | 40.8              |
>     | Multi-round DPO                                       | 23.8              |
>     | Multi-round DPO, with LoRA Proxy                      | 21.3              |
>     | Multi-round DPO, with LoRA Proxy, gDPO regularization | 16.3              |
>
>   - Multi-round DPO can achieve significant improvement compared to single-round DPO, with a total error rate reduction of about 17%. In multi-round DPO, the error rate will reduce by 2% if the LoRA proxy is used for training. Since the classical DPO loss might not be very stable during multi-round training, another 5% absolute error rate reduction is received by using gDPO in training.
>
> - **Weakness 5 and Question 5**
>
>   - Since the experimental resource consumption of mrDPO is very high, it is difficult to make comprehensive comparisons. However, we find that 5 or 6 rounds of mrDPO may be sufficient as the performance improvements are reduced, as shown by the "Proxy-gDPO" line and the "Direct-gDPO" line in Fig. 4.
>
>     Regarding the increased computational overhead of multiple LoRA proxies, it can be eliminated using an implementation trick. Specifically, we merge the LoRA module into the backbone LLM before each DPO round, as shown in Eqn. (10) in the paper. Therefore, in each round of DPO training, there is only one LoRA proxy hooked up to the LLM, and the memory and computational costs are always the same as using a single LoRA adaptor. During inference, we similarly merge the LoRA proxy into the model, which results in a model without any LoRA adaptor.
>
>     We train the model for 1k steps in each DPO round, which is enough for convergence. This takes about 6 hours in our experiments.
>
> - **Weakness 6 and Question 6**
>
>   - We do consider the synchronisation of audio and video in model training. Besides global video captioning, we also train our model to conduct local video captioning, which is to describe the paired audio and video events within a certain time interval. An example of local video captioning is shown below:
>
>     *Prompt*: Please describe the content of the video and audio from 00:11 to 00:38, focusing on the key events, visual elements, and sounds.
>
>     *Response*: From 00:11 to 00:38, the video continues to show the individual handling various pieces of paper and woodcuts, with the person explaining the history and significance of the woodcuts. The setting remains the same workshop environment with the ...
>
>     We also test our model using another internal benchmark,  which is an audio-visual QA test set focusing on audio-visual synchronization. There are mainly two types of questions in this benchmark:
>
>     - When the speaker mentions some specific things, what happens in the video?
>     - When some specific things happen in the video, what does the character says?
>
>      The accuracy of our model is shown below. Gemini-1.5's result is also shown:
>
>     |                 | Synchronized Audio-Visual QA Acc% |
>     | --------------- | --------------------------------- |
>     | Gemini-1.5-pro  | 88.9%                             |
>     | Video-SALMONN 2 | 65.8%                             |
>
>     Although the performance of our model on the Synchronized Audio-Visual QA is worse than Gemini-1.5-pro, the accuracy is still impressive considering its small model size and much smaller training data. Note that there is no other model to evaluate on this internal benchmark at the moment due to the lack of general speech and audio understanding ability of many public video LLMs. We will provide further case studies and will include the Synchronized Audio-Visual QA results in the revised paper.
>
> [1] Chai, Wenhao, et al. "AuroraCap: Efficient, Performant Video Detailed Captioning and a New Benchmark." arXiv preprint arXiv:2410.03051 (2024).

---

> > ### Author Response · Authors · 2024-11-23
> > **Response to Reviewer zUKj**
> >
> > We have uploaded the revised paper. The revised parts are marked blue.
> >
> > We are looking forward to your reply.

---

> > > ### Author Response · Authors · 2024-11-25
> > > **Response to Reviewer zUKj**
> > >
> > > Dear Reviewer zUKj,
> > >
> > > Thank you for your invaluable feedback on our paper. We submitted our detailed rebuttal a week ago, addressing all the concerns you raised. We greatly appreciate the time and effort you invest in the review process, and we would be grateful if you could take a moment to review our rebuttal.
> > >
> > > We hope our clarifications and revisions demonstrate the value and significance of our work. Thank you for considering our rebuttal, and we look forward to your feedback.
> > >
> > >
> > > Best regards,
> > >
> > > The Authors

---

> > > > ### Author Response · Authors · 2024-11-26
> > > > **Response to Reviewer zUKj**
> > > >
> > > > Dear Reviewer zUKj,
> > > >
> > > > Thank you for your invaluable feedback on our paper. We submitted our detailed rebuttal a week ago, addressing all the concerns you raised. We greatly appreciate the time and effort you invest in the review process, and we would be grateful if you could take a moment to review our rebuttal.
> > > >
> > > > We hope our clarifications and revisions demonstrate the value and significance of our work. Thank you for considering our rebuttal, and we look forward to your feedback.
> > > >
> > > > Best regards,
> > > >
> > > > The Authors

---

> > > > > ### Author Response · Authors · 2024-11-27
> > > > > **Response to Reviewer zUKj**
> > > > >
> > > > > Dear Reviewer zUKj,
> > > > >
> > > > > Thank you for your invaluable feedback on our paper. We submitted our detailed rebuttal a week ago, addressing all the concerns you raised. We greatly appreciate the time and effort you invest in the review process, and we would be grateful if you could take a moment to review our rebuttal.
> > > > >
> > > > > We hope our clarifications and revisions demonstrate the value and significance of our work. Thank you for considering our rebuttal, and we look forward to your feedback.
> > > > >
> > > > > Best regards,
> > > > >
> > > > > The Authors

---

> > ### Comment · Reviewer_zUKj · 2024-11-29
> > **Response to the authors**
> >
> > I sincerely apologize for my delayed response. I have been unwell over the past few days, which unfortunately caused a delay in getting back to you. Thank you for your understanding and patience, and I truly appreciate your consideration.
> >
> > **Weakness 1 and Question 1**
> > - Do you have any insights into why the performance declined after six rounds of DPO? I'm curious to understand if there's a specific reason behind it. The experiments look good, but I would like to know why the perfomance dropped.
> >
> > **Weakness 2 and Question 2**
> > - During prompt engineering, did you experiment with multiple prompts and evaluate their statistical significance?
> >
> > **Weakness 3 and Question 3**
> > My concern was addressed!
> >
> > **Weakness 4 and Question 4**
> > Although multi-round DPO with enhancements reduces the error rate significantly, the response does not address potential trade-offs, such as increased computational costs, longer training times, or challenges in convergence. **Could you estimate the computational overhead?** If possible.
> >
> > **Weakness 5 and Question 5**
> > - Discuss the generalizability of findings, particularly whether the observed performance improvements (5–6 rounds for convergence) hold across different tasks or datasets. I think I asked before. Additionally, I am curious too. Have you explored broader aspects of computational efficiency, including data handling and evaluation costs, to offer a more holistic view of mrDPO's practicality????
> >
> > **Weakness 6 and Question 6**
> > - As this is an internal benchmark and not open-source, we kindly request a detailed breakdown of the differences in training data, including aspects such as size, diversity, and annotation quality. Additionally, please provide an explanation of how these differences influence synchronization capabilities.
> > - Have you conducted error analysis to identify the root causes of Video-SALMONN 2’s lower performance (e.g., difficulties with specific types of cues or temporal misalignments)?
> > - Have you considered to try diverse audio-visual datasets or incorporating modules specifically designed for synchronization tasks?

---

> > > ### Author Response · Authors · 2024-11-29
> > > **Response to Reviewer zUKj (Part I)**
> > >
> > > Thank you for getting back to us! We're sorry to hear you haven't been feeling well and truly appreciate the time you've taken to review our paper and rebuttal. Below, we will respond to each of your points.
> > >
> > > **Weakness 1 and Question 1**
> > >
> > > We believe that this is mainly because in DPO training, only the video captioning task is trained, and therefore the newly introduced parameters (i.e. the LoRA proxies) in mrDPO mainly serve for more complete captioning. As a result, the model's QA abilities will decrease after mrDPO.
> > >
> > > We have carefully analysed the output of our model on Video-MME. We found that in many cases, the model is able to understand the question and actually has the answer. However, it picks the wrong answer, even though the reason it gives points to the correct answer.
> > >
> > > Here are some examples of bad cases we found in Video-MME:
> > >
> > > ---
> > >
> > > 1. https://www.youtube.com/watch?v=g1MCVp5xICM
> > >
> > > What is the right order of the tools appearing in the video when making a paper peony
> > >
> > > (a) Glue gun.    (b) Yellow paper.    (c) Pink paper.    (d) Scissors.
> > >
> > > A. (b)(a)(d)(c).    B. (b)(d)(a)(c).    C. (a)(b)(c)(d).    D. (d)(a)(c)(b).
> > >
> > > **Groundtruth:** **B**
> > >
> > > **Our model's output:** C. (b)(d)(a)(c).
> > >
> > >
> > >
> > > 2. https://www.youtube.com/watch?v=6DO8yOVYXr0
> > >
> > > What is the current score of the ongoing game?
> > >
> > > A. 3-2.    B. 2-3.    C. 2-2.    D. 3-3.
> > >
> > > **Groundtruth: A**
> > >
> > > **Our model's output:** B. The current score is tied at 3-2 in favor of Nebraska with Illinois leading the way in the first few frames.
> > >
> > > ---
> > >
> > > In these two cases, the model chooses the wrong options, but its descriptions point to the correct answers.
> > >
> > > The reason why rebirth tuning can mitigate this problem is that in rebirth tuning, lots of video QA training samples are used. Therefore, the model can restore the QA abilities.
> > >
> > >
> > >
> > > **Weakness 2 and Question 2**
> > >
> > > We have tried several prompts for GPT evaluation, and have found that as long as the prompts provide clear evaluation rules, the error of the evaluation results will be very small, within about ±2% of the total error rates.
> > >
> > > Besides, we have found that the error caused by different random seeds is small as well, which is also about ±2%. Therefore, in our evaluation pipeline, to get a more reliable result, we ask GPT3.5 seven times using different random seeds, and use the medium number as the evaluation results.
> > >
> > >
> > >
> > > **Weakness 4 and Question 4**
> > >
> > > For mrDPO training, it indeed requires large computational costs. The computational costs come from two parts: data generation and training. In one DPO round, for data generation (i.e. using the model of the previous DPO round to generate captioning samples), it takes about 1.5 hours to generate 26k video captions using 128 A100s, i.e. two captions per video. While for DPO training, it requires about 1000 steps of parameter updates for convergence using 64 A100s, which takes about 8 hours.
> > >
> > > Although the training process of mrDPO takes up a lot of computational resources, the inference cost of the final model actually remains completely unchanged. This is because the models always have the same structure in all training stages, meaning **no extra inference cost for deployment.** We observed no convergence issue with mrDPO.
> > >
> > >
> > >
> > > **Weakness 5 and Question 5**
> > >
> > > This question is challenging to verify, as there is currently no open-source training set with high-quality captions comparable to those used in video captioning. However, our training set is extensive, and the test set spans various video genres (as shown in Fig. 5), making the main findings and convergence stability theoretically reliable and generalizable. Regarding the hyperparameter choice of 5 to 6 epochs, it is just empirically selected and we believe identifying an appropriate number of epochs on another dataset would be straightforward, even if it differs from 5 or 6.
> > >
> > > Additionally, as our approach is specifically designed for video captioning, transferring it to other tasks may not be easy. For example, defining missing and hallucination rates is not straightforward for selection-based QA tasks.
> > >
> > > Regarding data handling, we have detailed the computational costs of mrDPO in Weakness 4 and Question 4. For evaluating the generated data, we used GPT-3.5, which is both cost-effective and efficient. Annotating 13,000 caption pairs took approximately 20 minutes. Furthermore, the resulting video-SALMONN 2 model can efficiently generate high-quality video captions for more data, helping to reduce data handling costs in practice. We will release the dataset used in this paper along with the final video-SALMONN 2 model checkpoint upon acceptance.

---

> > > > ### Author Response · Authors · 2024-11-29
> > > > **Response to Reviewer zUKj (Part II)**
> > > >
> > > > **Weakness 6 and Question 6**
> > > >
> > > > 1. For our training data for SFT, we have 13000 detailed video captions. Among them, 1500 captions are carefully labelled by humans, and the remains are generated by GPT-4o. All these videos contain rich audio information, especially human speech.
> > > >
> > > >    For human annotators, we ask them to carefully label synchronized audio-visual events, i.e. highlighting which audio events and video events are occurring at the same time.
> > > >
> > > >    To generate high-quality audio-visual captions using GPT-4o, we first cut the video into several 10-second segments. For each segment, we use SALMONN to label audio captions, Whisper to label speech transcriptions and GPT-4o to label pure visual captions. Finally, we integrate the audio captions, speech transcriptions and visual captions of all segments and ask GPT-4o to merge them into the final audio-visual caption of the video. The pipeline is described in Appendix C of the paper.
> > > >
> > > >    Based on the pipeline, in the final generated caption, 10-second granularity of the audio-visual synchronization can be guaranteed. Within 10 seconds, the audio-visual synchronization is mainly done by GPT-4o based on the audio caption and visual caption to be associated.
> > > >
> > > > 2. One possible reason for the lower synchronization performance is the varying sampling rate of visual frames in long videos. The model samples video frames at 1 FPS for videos less than 30 seconds, and uniformly samples 30 frames for videos longer than 30 seconds, while the entire audio sequence is fed into the audio encoder. This results in uncertainty in the synchronization of audio and visual frames, and confuses the model to some degree.
> > > >
> > > > 3. We are actively working on incorporating more diverse audio-visual datasets for synchronization tasks. For example, we have been gathering audio-visual speaker diarization datasets that require the model to jointly leverage both audio and video modalities. We believe that addressing audio-visual alignment is a crucial challenge and an important direction for future work related to this paper.

---

> ### Author Response · Authors · 2024-11-29
> **Response to Reviewer zUKj**
>
> Dear Reviewer zUKj,
>
> Thank you for your invaluable feedback on our paper. We submitted our detailed rebuttal two weeks ago, addressing all the concerns you raised. We greatly appreciate the time and effort you invest in the review process, and we would be grateful if you could take a moment to review our rebuttal.
>
> We hope our clarifications and revisions demonstrate the value and significance of our work. Thank you for considering our rebuttal, and we look forward to your feedback.
>
> Best regards,
>
> The Authors

---

### Author Response · Authors · 2024-12-03
**Official Comment by Authors**

**Dear Reviewers, Area Chairs and Senior Area Chairs,**

We thank all reviewers for their invaluable feedback on our work. We appreciate and highlight the positive feedback here:
- Our paper proposes a novel training pipeline (Reviewer zUKj, 7DpV and aYgd)
- Our paper focuses on detailed video captioning, which is a promising direction (Reviewer MeEe and aYgd)
- Our paper proposes novel metrics for detailed captioning evaluation (Reviewer zUKj and aYgd)
- Our approach achieved SOTA/competitive to SOTA performance on video captioning (Reviewer zUKj and MeEe)
- Our paper includes audio for video understanding (Reviewer zUKj and aYgd)
- The paper is clearly and well presented (Reviewer zUKj and 7DpV)

We have addressed all the questions and weaknesses the reviewers pointed out in our rebuttal. We have uploaded the revised version of the paper. Here is a summary:
- Evaluation of more video benchmarks (All reviewers)
  - We have evaluated our model on more video benchmarks in our rebuttal, including more video QA benchmarks (Video-MME, NeXT-QA, MVBench and VideoVista) and more video caption benchmarks (VDC). Reviewer aYgd and zUKj, the only reviewers who responded to us, both stated that this flaw in the paper has been perfectly addressed in the rebuttal.
- Improvement of the writing and illustrations (Reviewer MeEe and aYgd)
  - We have revised our paper to clarify more details, which has been acknowledged by Reviewer aYgd.
- Computational cost (Reviewer zUKj)
  - Although the training process of multi-round DPO takes up a lot of computational resources, the inference cost of the final model actually remains completely unchanged. This is because the models always have the same structure in all training stages, meaning no extra inference cost for deployment.
- Rationality of the new metrics (Reviewer zUKj and aYgd)
  - We have manually checked each step of computing the missing and hallucination rates, and the results are plausible. Besides, in our rebuttal, we conducted the ELO test following Reviewer aYgd's advice, and the results demonstrate that our proposed metrics are well aligned with human perception. This point is acknowledged by Reviewer aYgd.

We would like to summarize our contribution of this work:
- We develop video-SALMONN 2, a powerful audio-visual LLM that generates high-quality video captions, even surpassing GPT-4o and Gemini-1.5-Pro.
- We introduce a new evaluation pipeline that computes the missing and hallucination rates of audio-visual events in video captions.
- We propose novel training techniques to enhance detailed video captioning, including multi-round DPO with LoRA proxies and the guided loss function, and rebirth tuning.

While we have thoroughly addressed all the concerns raised by the reviewers, many of our responses have yet to receive further feedback. Our work represents a significant advancement in a challenging area, and we sincerely hope the reviewers can carefully review our rebuttal. We also kindly hope the Area Chair evaluates our work by taking into account its contribution and the detailed responses we have provided.

---

### Meta-Review · Area_Chair_KMXR · 2024-12-20

**Metareview:**

# Summary and Recommendation for Rejection

---

## Strengths:
1. **Innovative Training Techniques**:
   - Introduces **multi-round Directed Preference Optimization (mrDPO)** and **rebirth tuning** for optimizing detailed and accurate video captioning.
   - mrDPO stabilizes training through iterative updates and LoRA proxies, while rebirth tuning mitigates catastrophic forgetting in non-captioning tasks.

2. **Advanced Model Capabilities**:
   - The proposed model, **video-SALMONN 2**, significantly improves video captioning tasks, surpassing GPT-4o and Gemini-1.5-Pro in custom benchmarks.
   - Synchronized audio-visual inputs enhance the model's multimodal understanding, producing richer and more contextually accurate descriptions.

3. **Novel Evaluation Metrics**:
   - Proposes new metrics for video captioning, including missing and hallucination rates.

4. **Comprehensive Experimental Validation**:
   - Demonstrates significant reductions in error and repetition rates, supported by thorough ablation studies on mrDPO and LoRA proxies.

5. **Potential Community Impact**:
   - The authors commit to releasing datasets, code, and model checkpoints, fostering transparency and future research opportunities.

---

## Weaknesses:
1. **Limited External Benchmarking**:
   - Evaluation is predominantly on internal datasets, with limited testing on public benchmarks (e.g., VideoMME Short, VDC). This raises concerns about the generalizability of the results.

2. **High Computational Costs**:
   - mrDPO’s iterative training process requires significant computational resources, limiting its accessibility to researchers with fewer resources.
   - Insufficient quantitative analysis of the trade-offs between performance gains and computational overhead.

3. **Dependence on Other Models for Evaluation**:
   - Metrics like hallucination and missing rates rely on GPT-3.5/4 for evaluation, introducing potential biases and reproducibility challenges.

4. **Unclear Advantage of mrDPO**:
   - Although mrDPO reduces error rates, it introduces issues like unnatural text patterns, which require additional fine-tuning. The advantage over single-round DPO remains unclear due to limited direct comparisons.

5. **Incomplete Presentation**:
   - Methodological details are not fully explained, such as specific loss functions, definitions of key terms, and the process of sampling captions.
   - Missing qualitative examples that showcase the incremental improvements of mrDPO and rebirth tuning.

---

## Authors’ Mitigation:
1. **Expanded Benchmarks**:
   - Added evaluations on new video QA datasets (e.g., NeXT-QA, MVBench, VideoVista) and public video captioning benchmarks (e.g., VDC).

2. **Improved Clarity**:
   - Revised the manuscript to clarify methodology, including detailed explanations of training losses, caption sampling, and mrDPO dynamics.

3. **Addressed Computational Concerns**:
   - Quantified computational costs for data generation and training, explaining methods to reduce overhead using merging techniques for LoRA proxies.

4. **Evaluated Metrics**:
   - Justified the reliability of new metrics through ELO tests and manual validations, addressing concerns about potential biases.

---

## Remaining Weaknesses:
1. **Generality and Scalability**:
   - Broader validation on open benchmarks like MSR-VTT and VATEX remains missing, hindering general applicability.
   - Computational demands of mrDPO make the approach less feasible for researchers with limited resources.

2. **Evaluation Pipeline Limitations**:
   - Heavy reliance on external LLMs (GPT-3.5/4) for evaluation raises concerns about reproducibility and objectivity.

3. **Incomplete Analysis**:
   - The paper lacks a detailed breakdown of individual components of mrDPO and their specific contributions to performance improvements.

4. **Presentation Issues**:
   - Despite revisions, some sections still lack sufficient detail, limiting full reproducibility.

---

## Recommendation:  **Reject**

### **Justification**:
1. **Evaluation Shortcomings**:
   - Over-reliance on internal datasets and limited external benchmarking weaken the paper’s generalizability and broader applicability.

2. **High Computational Overhead**:
   - The computational demands of mrDPO, combined with insufficient analysis of trade-offs, limit the practicality of the proposed approach.

3. **Insufficient Clarity and Reproducibility**:
   - Key methodological details remain unclear, and dependence on external LLMs for evaluation introduces significant reproducibility concerns.

4. **Potential but Unproven Impact**:
   - While promising, the paper does not yet demonstrate sufficient breadth or robustness to meet the standards of a top-tier conference like ICLR.

The paper would benefit from more comprehensive external benchmarking, improved computational efficiency analyses, and clearer methodological explanations to better establish its contributions.

**Additional Comments On Reviewer Discussion:**

Please refer to details in the above section

---

### Decision · Program_Chairs · 2025-01-22

Reject